# Multi-SWE-bench: A Multilingual Benchmark for Issue Resolving

**Daoguang Zan**[*†]   **Zhirong Huang**[*]   **Wei Liu**[*]   **Hanwu Chen**   **Shulin Xin**
**Linhao Zhang**   **Qi Liu**   **Aoyan Li**   **Lu Chen**   **Xiaojian Zhong**   **Siyao Liu**
**Yongsheng Xiao**   **Liangqiang Chen**   **Yuyu Zhang**   **Jing Su**
**Tianyu Liu**   **Rui Long**   **Ming Ding**[†]   **Liang Xiang**
[*]Equal contribution.   [†]Corresponding author.
ByteDance Seed
{zandaoguang, shen.kai}@bytedance.com

## Abstract

The task of issue resolving aims to modify a codebase to generate a patch that addresses a given issue. However, most existing benchmarks focus almost exclusively on Python, making them insufficient for evaluating Large Language Models (LLMs) across different programming languages. To bridge this gap, we introduce a multilingual issue-resolving benchmark, called Multi-SWE-bench, covering 8 widely used programming languages: Python, Java, TypeScript, JavaScript, Go, Rust, C, and C++. In particular, this benchmark includes a total of $2,132$ high-quality instances, carefully curated by $68$ expert annotators, ensuring a reliable and accurate evaluation of LLMs on the issue-resolving task. Based on human-annotated results, the issues are further classified into three difficulty levels. We evaluate a series of state-of-the-art models on Multi-SWE-bench, utilizing both procedural and agent-based frameworks for issue resolving. Experimental results based on Multi-SWE-bench reveal three key findings: (1) *Limited generalization across languages*: While existing LLMs perform well on Python issues, their ability to generalize across other languages remains limited; (2) *Performance aligned with human-annotated difficulty*: LLM-based agents' performance closely aligns with human-assigned difficulty, with resolved rates notably decreasing as issue complexity rises; and (3) *Performance drop on cross-file issues*: The performance of current methods significantly deteriorates when handling cross-file issues. These findings highlight the limitations of current LLMs and underscore the need for more robust models capable of handling a broader range of programming languages and complex issue scenarios.

## 1   Introduction

Automating software engineering tasks with large language models (LLMs) has gained considerable attention [45, 51, 19, 18] recently. Beyond code generation, the issue resolving task proposed by SWE-bench [20] changes the role of LLMs from code assistants to fully autonomous AI programmers. SWE-bench contains $2,294$ issues from $12$ widely-used open-sourced Python libraries. LLMs are tasked to generate a patch based on the issue description along with the buggy code repository. SWE-bench Verified is a subset of $500$ human-validated issues selected from SWE-bench, chosen for appropriately scoped unit tests and well-specified issue descriptions. Within less than one year, the resolving rate on SWE-bench Verified increased from $0.40\%$ [20] (for RAG+GPT3.5) to $65.40\%$ [6] (for Augment Agent v0).

39th Conference on Neural Information Processing Systems (NeurIPS 2025) Track on Datasets and Benchmarks.

Although existing works based on SWE-bench demonstrate significant progress in Python-based issue resolving, the diversity of programming languages in real-world repositories presents additional challenges that remain unexplored. In particular, repositories in different languages follow distinct programming paradigms, idiomatic patterns, and runtime behaviors, which may impact the effectiveness of current approaches. This raises the question of whether the impressive performance of existing agents on Python issues can be generalized to other widely used languages, such as Java, TypeScript, JavaScript, Go, Rust, C, and C++.

To answer this question, we introduce Multi-SWE-bench, a multilingual benchmark for issue resolving, consisting of $2,132$ issues across $8$ widely used programming languages: Python, Java, TypeScript, JavaScript, Go, Rust, C, and C++. To construct a reliable benchmark for evaluating the ability of agents to resolve real-world software issues, we employ a systematic five-phase pipeline. First, we select high-quality repositories from GitHub based on star ratings and runnability counts to ensure both popularity and practical usability. Second, we collect issue-related pull requests (PRs) along with their corresponding metadata. Third, we build Dockerized environments for each PR by extracting dependencies from CI/CD workflows and documentation to ensure reproducible execution. Fourth, we validate PRs by analyzing test outcomes across patch configurations, retaining only those with clear bug-fixing effects and no regressions. Fifth, we perform rigorous manual verification through dual annotation and cross-review, ensuring high-quality ground truth aligned with SWE-bench verified standards. By ensuring diversity, executability, and human-verified correctness, Multi-SWE-bench sets a high standard for evaluating LLMs on realistic and non-trivial issue-resolving tasks.

With its wide coverage of languages and issue types, Multi-SWE-bench introduces realistic challenges that push the boundaries of LLM-based software agents. Specifically, we use Multi-SWE-bench to evaluate the generalizability of 3 representative methods (i.e., Agentless [39], SWE-agent [40], and OpenHands +CodeAct v2.1 [37]) based on 12 top-performing models. Our evaluation provides a comparative analysis of the overall effectiveness of these methods across eight programming languages, offering insights into their cross-language capabilities. Furthermore, we conduct a fine-grained analysis of the key factors influencing model performance and investigate failure cases for each language to identify underlying challenges and limitations. Through comprehensive analysis and comparison, we provide a good understanding of existing models and shed light on future directions and further progress. For example, our findings show that models perform generally better when issue descriptions are longer, indicating a strong reliance on rich contextual grounding; in contrast, resolved rates drop sharply when fix patches exceed 600 tokens or touch more than one file, exposing weaknesses in long-context retention and cross-file reasoning. These findings aim to delineate the current boundary of LLM capabilities in software engineering and reveal the key challenges to real-world deployment.

In summary, our main contributions are: (1) Multi-SWE-bench, a multilingual issue resolving benchmark with 2,132 human-validated GitHub issues across 8 widely used programming languages; (2) A large-scale evaluation of 12 state-of-the-art LLMs based on 3 representative methods (i.e., Agentless, SWE-agent, OpenHands) on Multi-SWE-bench, comparing performance across eight programming languages and revealing biases across models and methods; (3) Fully open-sourcing the benchmark, code, and Docker images to support community growth and advance research.[1]

## 2 Related Work

The remarkable performance of LLMs in code-related tasks has motivated substantial research to study their role in automating software engineering. To evaluate the capabilities and limitations of existing approaches, a wide range of benchmarks for code-related tasks has been developed. Early efforts in this domain focused on primarily evaluating models in monolingual program-level evaluations [3, 31, 16, 9, 7, 38]. As LLMs advanced, benchmarks evolved in two key dimensions to better align with real-world software engineering scenarios. First, benchmarks shift from monolingual to multilingual tasks, with growing interest and practical needs in evaluating LLMs' performance across multiple programming languages. Examples include Multilingual-HumanEval [5] and HumanEval-X [50], which extend the HumanEval [8] benchmark to multiple languages, and MBXP [4], which extends MBPP to multilingual scenarios. Second, benchmarks shift from program-level to repository-level

---

[1]All open-source resources can be accessed through `https://multi-swe-bench.github.io`.

Table 1: A comparison of Multi-SWE-bench to existing issue resolving benchmarks. Multi-SWE-bench distinguishes itself by (1) covering a broad range of programming languages, (2) filtering out problematic issues by human verification, and (3) offering a well-defined difficulty stratification (Diff. Strat.), enabling a more realistic, reliable, and systematic evaluation of the capabilities of LLMs.

| Benchmarks | Published Date | Programming Languages | #Issues | #Repos | Verified | Diff. Strat. |
|---|---|---|---|---|---|---|
| SWE-bench [20] | 2023/10/10 | Python | 2294 | 12 | ✗ | ✗ |
| SWE-bench Verified [20] | 2024/08/13 | Python | 500 | 12 | ✓ | ✗ |
| SWE-bench Multimodal [41] | 2024/10/04 | JavaScript | 617 | 17 | ✓ | ✗ |
| Visual SWE-bench [48] | 2024/12/23 | Python | 133 | 11 | ✓ | ✗ |
| SWE-Lancer [25] | 2025/02/17 | TypeScript, JavaScript | 1488 | 1 | ✓ | ✗ |
| SWE-PolyBench[†] [30] | 2025/04/17 | Python, Java, TypeScript, JavaScript | 2110 | 21 | ✗ | ✗ |
| SWE-bench Multilingual[†] [1] | 2025/05/06 | Java, TypeScript, JavaScript, Go, Rust, C, C++, PHP, Ruby | 300 | 42 | ✗ | ✗ |
| OmniGIRL[†] [13] | 2025/05/07 | Python, Java, TypeScript, JavaScript | 959 | 15 | ✗ | ✗ |
| Multi-SWE-bench (Ours) | 2025/04/03 | Python, Java, TypeScript, JavaScript, Go, Rust, C, C++ | 1632 | 39 | ✓ | ✓ |

Note: [†] indicates benchmarks released released soon after Multi-SWE-bench, representing concurrent work.

tasks, focusing on more complex scenarios such as library-oriented code generation [44], repository-level code completion [47, 23, 10, 24, 42], and bug fix [26, 28, 32]. These evolving benchmarks aim to provide a more comprehensive evaluation of LLMs in real-world scenarios.

In addition to existing benchmarks, SWE-bench [20] has gained significant attention since its release. Instead of focusing on isolating code subtasks into separate datasets, SWE-bench addresses a broader range of tasks through repository-level issue resolving. These issue resolving tasks, including bug fixing, new feature requests, and optimization, which provide a more comprehensive evaluation of LLMs' ability to automating software development. However, some issues in SWE-bench have underspecified descriptions or overly specific and irrelevant tests. To address this, SWE-bench Verified [20] filters out these issues by a questionnaire-based human verification process, creating a refined subset consisting of solvable and testable issues. While SWE-bench is limited to textual context, SWE-bench Multimodal [41] and Visual SWE-bench [48] extend evaluation to systems fixing bugs in visually-oriented and user-facing applications. SWE-Lancer [25] focuses on JavaScript and TypeScript, featuring over $1,400$ freelance tasks from Upwork, including technical and managerial tasks. Despite these advancements, the performance of LLMs on other widely used programming languages remains underexplored. Since the release of our Multi-SWE-bench, several multilingual benchmarks have been introduced, including SWE-PolyBench [30], SWE-bench Multilingual [1], and OmniGIRL [13]. Multi-SWE-bench distinguishes itself by (1) covering $2,132$ GitHub issues across 8 widely used languages, (2) ensuring high-quality issues through manual verification by 68 expert annotators, and (3) offering a well-defined difficulty stratification framework, thus ensuring a more realistic, reliable, and systematic evaluation of LLMs' capabilities.

# 3 Multi-SWE-bench Construction

To evaluate the generalizability of LLMs as issue resolvers, eight widely used programming languages are selected to construct Multi-SWE-bench through five phases. As shown in Fig. 1, the first four phases create a large pool of candidate data for each language, while the fifth phase finalizes the Multi-SWE-bench through manual verification.

## 3.1 Phase 1: Repository Selection

We carefully curate a diverse set of high-quality GitHub repositories for each of the eight target programming languages. The selection process is guided by the following criteria: (1) Popularity and Maintenance: Repositories must have over 500 GitHub stars and demonstrate active maintenance for at least six months. In addition, we prioritize repositories frequently recommended in Google searches using keywords such as "high-quality", "well-maintained", and "popular". (2) CI/CD Support: Selected repositories are required to include CI/CD configurations (e.g., workflows under `.github/workflows/`) to ensure automated testing and reproducibility. (3) Build Viability: After minimal manual setup, the latest commit must be buildable and testable in a clean environment, ensuring compatibility with modern tooling and infrastructure.

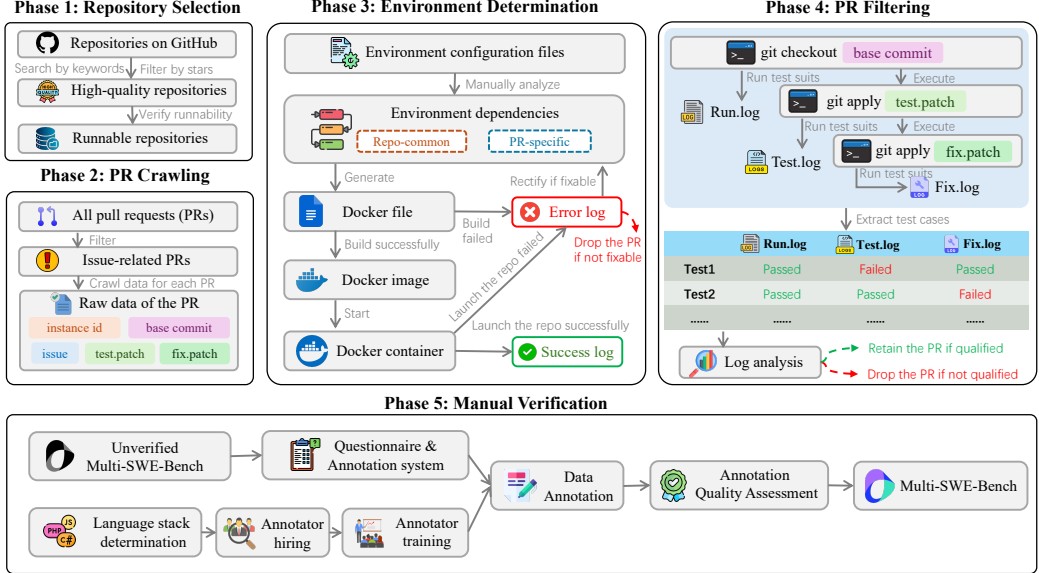

Figure 1: Construction of Multi-SWE-bench.

## 3.2 Phase 2: Pull Request Crawling

This phase aims to crawl issue-resolving pull requests (PRs) for each repository selected in phase 1. All PRs from the repository are collected and then filtered based on the following criteria:

- Linked with at least one GitHub issue: The PR must be linked to at least one issue to ensure it addresses a clearly defined bug report or feature request.

- Modified test files: The PR must include changes to test files, guaranteeing proper testing is in place to verify the correctness of the fix patches.

- Merged into the main branch: The PR must be merged into the main branch, indicating it has been accepted by the repository's maintainers and fully integrated.

After filtering, detailed information is gathered for each PR, including attributes such as issue description, base commit, fix.patch, and test.patch.

## 3.3 Phase 3: Environment Determination

To ensure a faithful execution and evaluation, each pull request (PR) must be reproducibly built and run in an isolated environment. In this phase, we achieve this by creating a Docker-based runtime for each PR by automatically identifying and provisioning its dependencies. The process begins with a manual review of environment-related artifacts, such as CI/CD configuration files (e.g., GitHub Actions), repository documentation (e.g., README files), and exploratory trial runs. From this analysis, we categorize dependencies into two types: repo-common dependencies (shared across the repository) and PR-specific dependencies (introduced or modified by the target PR).

Using the extracted dependency information, we generate a tailored Dockerfile and build the corresponding Docker image. If the build fails, we analyze error logs to identify issues like missing dependencies or version conflicts and iteratively update the Dockerfile or supporting scripts. If the errors are unresolvable, we discard the PR. Once the image builds successfully, we verify that the repository launches correctly at the specified commit, ensuring all services and configurations are functional. If the launch fails, corrective actions are taken; if successful, we obtain a validated, executable container for downstream evaluation. This process ensures a reliable, functional environment for testing and analysis.

Table 2: Statistics of the Multi-SWE-bench (excluding Python). #A2P2P, #A2F2P, and #A2N2P represent the average counts of Any→PASSED&FAILED&NONE→PASSED unit tests.

| Repository | | | Instance | Issue description | Fix patches | | | Unit tests | | |
|---|---|---|---|---|---|---|---|---|---|---|
| Org/Repo | #Files | #LoC | #Num | Avg. #Tokens | Avg. #Lines | Avg. #Hunks | Avg. #Files | #A2P2P | #A2F2P | #A2N2P |
| Java | | | | | | | | | | |
| alibaba/fastjson2 | 4244 | 443.8k | 6 | 459.2 | 10.5 | 1.3 | 1.2 | 1243.5 | 0.8 | 1020.5 |
| elastic/logstash | 562 | 59.9k | 38 | 1600.4 | 212.3 | 10.0 | 4.6 | 554.7 | 1.9 | 256.2 |
| mockito/mockito | 986 | 84.0k | 6 | 315.2 | 92.5 | 10.3 | 4.7 | 97.2 | 1.0 | 3.8 |
| apache/dubbo | 3939 | 402.1k | 3 | 774.0 | 9.3 | 3.0 | 1.3 | 2.0 | 57.0 | 0.0 |
| fasterxml/j-core | 366 | 105.7k | 18 | 304.7 | 33.8 | 4.8 | 2.1 | 2.0 | 85.6 | 0.0 |
| fasterxml/j-dbind | 1230 | 217.5k | 42 | 621.5 | 35.1 | 3.9 | 2.1 | 2.0 | 73.8 | 0.0 |
| fasterxml/j-dfmt-xml | 206 | 23.0k | 5 | 1071.8 | 98.4 | 10.4 | 3.2 | 2.0 | 94.2 | 0.0 |
| google/gson | 261 | 48.0k | 5 | 365.8 | 35.8 | 4.6 | 1.8 | 2.0 | 62.6 | 0.0 |
| google-ct/jib | 604 | 75.5k | 5 | 1094.6 | 15.2 | 3.2 | 2.6 | 2.0 | 96.2 | 0.0 |
| TypeScript | | | | | | | | | | |
| darkreader/darkreader | 189 | 26.2k | 2 | 749.5 | 13.0 | 2.0 | 1.5 | 41.0 | 3.5 | 0.0 |
| mui/material-ui | 27632 | 698.6k | 174 | 508.6 | 331.2 | 20.2 | 12.0 | 5001.3 | 2.3 | 836.8 |
| vuejs/core | 509 | 128.2k | 48 | 694.8 | 22.9 | 3.5 | 1.9 | 2920.4 | 3.0 | 0.0 |
| JavaScript | | | | | | | | | | |
| ag/gh-rdme-stats | 69 | 11.8k | 19 | 287.1 | 123.6 | 13.5 | 4.8 | 108.9 | 3.5 | 3.4 |
| axios/axios | 166 | 21.0k | 4 | 490.8 | 179.5 | 7.8 | 4.0 | 68.5 | 1.2 | 0.0 |
| expressjs/express | 142 | 17.3k | 4 | 177.5 | 7.2 | 2.2 | 1.5 | 808.2 | 1.5 | 65.2 |
| iamkun/dayjs | 324 | 17.1k | 56 | 325.6 | 21.7 | 2.7 | 2.0 | 60.4 | 1.2 | 3.2 |
| Kong/insomnia | 526 | 182.0k | 1 | 709.0 | 1.0 | 1.0 | 1.0 | 105.0 | 1.0 | 0.0 |
| sveltejs/svelte | 2800 | 105.9k | 272 | 618.9 | 72.0 | 8.4 | 4.0 | 4904.2 | 5.5 | 0.0 |
| Go | | | | | | | | | | |
| cli/cli | 737 | 165.1k | 397 | 347.6 | 103.8 | 9.0 | 3.9 | 1997.0 | 2.9 | 31.0 |
| grpc/grpc-go | 981 | 260.8k | 16 | 276.1 | 81.8 | 7.7 | 2.8 | 230.4 | 0.6 | 6.6 |
| zeromicro/go-zero | 960 | 117.6k | 15 | 205.2 | 52.4 | 4.9 | 2.7 | 1318.9 | 0.3 | 43.9 |
| Rust | | | | | | | | | | |
| BurntSushi/ripgrep | 98 | 45.4k | 14 | 553.7 | 1604.9 | 21.9 | 7.5 | 233.2 | 1.1 | 8.1 |
| clap-rs/clap | 321 | 70.4k | 132 | 987.0 | 147.1 | 15.7 | 4.7 | 489.5 | 3.1 | 378.8 |
| nushell/nushell | 1479 | 264.2k | 14 | 795.6 | 155.0 | 10.6 | 4.3 | 798.6 | 2.6 | 336.6 |
| rayon-rs/rayon | 191 | 36.9k | 2 | 153.5 | 637.5 | 5.5 | 2.0 | 113.5 | 0.5 | 171.0 |
| serde-rs/serde | 188 | 36.5k | 2 | 171.5 | 72.5 | 3.0 | 3.0 | 0.0 | 0.0 | 294.5 |
| sharkdp/bat | 83 | 22.0k | 10 | 638.2 | 239.5 | 14.1 | 5.9 | 152.7 | 1.7 | 33.6 |
| sharkdp/fd | 24 | 6.7k | 14 | 167.8 | 55.8 | 7.8 | 4.5 | 186.5 | 1.1 | 0.0 |
| tokio-rs/bytes | 33 | 11.9k | 5 | 188.0 | 45.0 | 5.6 | 1.8 | 23.2 | 0.4 | 91.6 |
| tokio-rs/tokio | 727 | 141.5k | 25 | 590.0 | 139.8 | 10.6 | 3.5 | 26.6 | 0.0 | 287.4 |
| tokio-rs/tracing | 241 | 60.9k | 21 | 472.0 | 597.2 | 39.3 | 7.1 | 30.8 | 0.2 | 182.0 |
| C | | | | | | | | | | |
| facebook/zstd | 276 | 119.8k | 29 | 496.6 | 67.6 | 10.9 | 3.0 | 0.8 | 0.5 | 5.6 |
| jqlang/jq | 80 | 43.0k | 17 | 429.8 | 26.1 | 2.7 | 1.8 | 27.2 | 1.0 | 0.1 |
| ponylang/ponyc | 285 | 80.2k | 82 | 480.2 | 205.4 | 15.6 | 5.7 | 997.6 | 1.9 | 388.8 |
| C++ | | | | | | | | | | |
| catchorg/Catch2 | 399 | 58.0k | 12 | 357.3 | 469.0 | 15.4 | 8.2 | 19.9 | 0.7 | 17.6 |
| fmtlib/fmt | 25 | 36.4k | 41 | 397.7 | 36.8 | 3.0 | 1.1 | 9.3 | 0.0 | 9.3 |
| nlohmann/json | 477 | 124.7k | 55 | 905.5 | 405.8 | 27.9 | 6.5 | 26.5 | 0.0 | 42.9 |
| simdjson/simdjson | 455 | 229.7k | 20 | 320.2 | 768.5 | 35.5 | 11.0 | 18.6 | 0.0 | 41.5 |
| yhirose/cpp-httplib | 33 | 50.9k | 1 | 240.0 | 1.0 | 1.0 | 1.0 | 272.0 | 1.0 | 0.0 |

## 3.4 Phase 4: Pull Request Filtering

In this phase, we perform a semantic validation to ensure each PR obtained from previous phase meets the requirements of issue resolving. This is done by analyzing test behaviors under controlled patch configurations. For each PR, unlike SWE-bench which runs only relevant tests, we run the full test suit under the following three settings: (1) `Run.log`: Tests are executed on the base commit; (2) `Test.log`: The test.patch is applied to the base commit before execution; (3) `Fix.log`: Both the test.patch and the fix.patch are applied to the base commit before execution.

Based on these logs, we extract the execution status of each test case. Each test case is summarized by its status transition across the three settings. For instance, a test case with PASSED, FAILED, and PASSED statuses in run.log, test.log, and fix.log, respectively, is represented as PASSED→FAILED→PASSED. We apply the following filtering rules to determine eligible PRs:

- PRs with any ANY→PASSED→FAILED transitions are discarded to ensure that no potential regressions are introduced by the fix.patch.

- PRs without at least one ANY→FAILED→PASSED transition are discarded, as they do not demonstrate any effective bug fix.

- PRs exhibiting abnormal transitions such as PASSED→NONE/SKIPPED→FAILED are discarded to eliminate ambiguous test behaviors.

After applying these criteria, we retain $2,456$ issue-resolving instances. For each instance, we extract test cases exhibiting transitions of the form Any→FAILED/PASSED/SKIPPED/NONE→PASSED, and include them in the dataset to enable fine-grained and reliable evaluation.

### 3.5 Phase 5: Manual Verification

To ensure the reliability of Multi-SWE-bench, we conduct manual verification on the $2,456$ issue-resolving instances. Our verification process follows the annotation guidelines of the recently released SWE-bench-verified[2]. In detail, we recruit $68$ annotators through outsourcing, and all annotators have at least two years of experience in the target language and a relevant bachelor's degree or higher.

Before annotation, each annotator undergoes training covering the task's background, objectives, procedures, deliverables, and quality standards. To ensure consistency and accuracy, we establish real-time discussion channels to provide guidance and address edge cases collaboratively. Each instance is independently labeled by two annotators. Afterward, the annotations are cross-reviewed to reach a final, agreed-upon label. To maintain high quality, a dedicated internal team of 14 experienced engineers assesses the annotations, producing reference answers and verifying that outsourced annotations meet an $80\%$ accuracy threshold. After thorough manual verification, 1,632 high-quality instances covering 7 languages other than Python are retained as the final dataset, filtered according to specific criteria outlined in the verification questionnaire[3]: Q2.1=0 & Q3.1$\in$\{2,3\} & Q4.1$\in$\{2,3\}. The annotation details can be found in Appendix A. All annotation results are publicly available to ensure dataset transparency. Together with the $500$ Python instances from SWE-bench Verified, our final Multi-SWE-bench consists of 2,132 instances.

To further advance research on issue resolving, we also introduce two complementary resources: (1) *Multi-SWE-bench Mini* (see Appendix E.1), a lightweight subset of Multi-SWE-bench designed to enable faster and more cost-effective evaluation; (2) *Multi-SWE-RL* (see Appendix B), an open-source community aiming at creating large-scale reinforcement learning (RL) training datasets. Moreover, we summarize the troubleshooting encountered during the dataset construction process in Appendix C.

## 4 Characteristics of Multi-SWE-bench

**Overall statistics of Multi-SWE-bench.** Tab. 2 presents an overview of the key statistics of Multi-SWE-bench. Specifically, it includes $2,132$ issue-resolving instances, spanning $8$ popular languages: Python, Java, TypeScript (TS), JavaScript (JS), Go, Rust, C, and C++. These repositories vary significantly in size and complexity, with the number of files ranging from $24$ to $27.6$k, and lines of code from $6.7$k to $698.6$k. Similarly, patch complexity also differs across repositories and languages. Rust and C++ projects frequently require large-scale edits, with some instances modifying over $200$ lines and 7 files per patch (e.g., `BurntSushi/ripgrep` and `simdjson/simdjson`). Conversely, TS and JS patches tend to be more localized and atomic, often involving under 3 hunks and fewer than 2 files. Moreover, all repositories come with strong test coverage, providing reliable signals for verifying patch correctness, as confirmed by the manual verification in Sec. 3.5.

**Difficulty stratification.** In Multi-SWE-bench, we adopt a human-aligned, time-based difficulty stratification approach to systematically evaluating the capabilities of LLMs. Specially, in the manual verification phase, each issue is annotated by human that can be resolved into one of four time intervals: $\leq$15 minutes, 15 minutes–1 hour, 1–4 hours, and $\geq$4 hours. Based on these time estimates, we further define three levels of difficulty: easy ($\leq$15 minutes), medium (15 minutes–1 hour), and hard ($\geq$1 hour). Tab. 3 summarizes the distribution of difficulty levels across different programming languages. We observe clear trends across these categories: As difficulty increases, issues tend to have longer descriptions, and the corresponding patches involve more lines, hunks, and files. Such categorization provides a more accurate and human-aligned measure of problem difficulty.

## 5 Experimental Setups

**Methods.** We evaluate three representative methods in our experiments, covering both procedural and agent-based frameworks: Agentless [39], SWE-agent [40], and OpenHands + CodeAct v2.1 [37]. These methods are initially designed for Python. We extended them to support the multilingual environment of Multi-SWE-bench, forming MagentLess, MSWE-agent, and MopenHands, respec-

---

[2]`https://openai.com/index/introducing-swe-bench-verified`
[3]`https://github.com/multi-swe-bench/multi-swe-bench/blob/main/docs/manual-verification/questionnaire-demo.pdf`

Table 3: Distribution of Multi-SWE-bench instances by difficulty and language.

| Language | Difficulty | Instance #Num | Issue description Avg. #Tokens | Fix patches | | | Unit tests | | |
|---|---|---|---|---|---|---|---|---|---|
| | | | | Avg. #Lines | Avg. #Hunks | Avg. #Files | #A2P2P | #A2F2P | #A2N2P |
| Python | Easy | 194 | 417.9 | 5.0 | 1.4 | 1.0 | 116.2 | 3.9 | 0 |
| | Medium | 261 | 555.9 | 14.1 | 2.5 | 1.3 | 115.4 | 2.4 | 0 |
| | Hard | 45 | 589.8 | 55.8 | 6.8 | 2.0 | 166.3 | 2.9 | 0 |
| Java | Easy | 27 | 733.8 | 12.4 | 2.6 | 1.8 | 126.8 | 58.0 | 76.1 |
| | Medium | 65 | 843.3 | 36.2 | 4.6 | 2.1 | 182.3 | 58.6 | 136.9 |
| | Hard | 36 | 1039.0 | 246.1 | 11.9 | 5.4 | 389.1 | 21.8 | 136.9 |
| TypeScript | Easy | 72 | 600.1 | 8.3 | 2.1 | 1.5 | 4806.8 | 2.0 | 0.0 |
| | Medium | 88 | 566.9 | 74.3 | 8.8 | 4.3 | 4854.6 | 2.8 | 214.3 |
| | Hard | 64 | 472.8 | 806.6 | 43.2 | 26.5 | 3706.1 | 2.7 | 1980.4 |
| JavaScript | Easy | 10 | 282.4 | 4.7 | 1.8 | 1.6 | 616.8 | 1.2 | 35.1 |
| | Medium | 105 | 505.8 | 15.5 | 2.6 | 2.1 | 3161.0 | 3.6 | 0.8 |
| | Hard | 241 | 578.7 | 92.2 | 10.1 | 4.5 | 4169.9 | 5.2 | 0.3 |
| Go | Easy | 141 | 411.7 | 26.6 | 4.0 | 2.7 | 2181.0 | 2.6 | 20.4 |
| | Medium | 153 | 331.4 | 49.6 | 6.9 | 2.6 | 1832.5 | 2.2 | 25.7 |
| | Hard | 134 | 274.0 | 238.6 | 16.0 | 6.6 | 1704.2 | 3.4 | 46.7 |
| Rust | Easy | 66 | 808.2 | 318.7 | 7.0 | 3.3 | 465.2 | 3.2 | 212.0 |
| | Medium | 126 | 814.7 | 113.6 | 10.6 | 3.7 | 343.0 | 1.8 | 300.5 |
| | Hard | 47 | 599.4 | 629.0 | 45.2 | 10.3 | 232.3 | 1.1 | 334.0 |
| C | Easy | 30 | 551.4 | 16.4 | 3.7 | 2.2 | 424.8 | 0.8 | 208.2 |
| | Medium | 54 | 449.9 | 36.7 | 5.5 | 2.5 | 715.5 | 1.0 | 228.2 |
| | Hard | 44 | 460.2 | 381.1 | 28.0 | 8.7 | 702.5 | 2.4 | 306.3 |
| C++ | Easy | 28 | 494.5 | 25.2 | 4.4 | 2.2 | 45.0 | 0.1 | 15.7 |
| | Medium | 59 | 427.5 | 204.2 | 7.6 | 3.3 | 18.2 | 0.1 | 23.0 |
| | Hard | 42 | 904.2 | 763.7 | 47.2 | 11.1 | 9.3 | 0.0 | 47.2 |

tively. We systematically adapted the aforementioned methods to the multilingual setting, with details provided in Appendix D.

**LLMs.** Experiments use 12 representative LLMs: GPT-4o (gpt-4o-2024-11-20), OpenAI-o1 (o1-2024-12-17), OpenAI-o3-mini-high (o3-mini-2025-01-31 high), Claude-3.5-Sonnet (claude-3-5-sonnet-20241022), Claude-3.7-Sonnet (claude-3-7-sonnet-20250219), DeepSeek-V3, DeepSeek-R1, Qwen2.5-72B-Instruct, Doubao-1.5-pro, Doubao-1.5-thinking, Gemini-2.5-Pro, and Llama-4-Maverick.

**Metrics.** Following prior work [20, 49, 39], we report the following primary evaluation metrics for end-to-end performance: (1) Resolved Rate (%): the percentage of issues resolved. (2) Avg. Cost ($): the average cost per issue. Additionally, we also provide an analysis of issue location accuracy in Appendix E.2 and report % Success Location, which is defined as a patch that contains a correct location if it modifies a superset of all locations in the ground truth fix patch.

# 6 Experimental Results

## 6.1 Performance on Multi-SWE-bench

**Limited generalization beyond Python.** From Tab. 4, it can be observed that existing methods demonstrate strong performance in resolving Python issues but struggle to generalize effectively across other languages. For example, LLMs such as OpenAI-o1 achieve high resolved rates for Python but significantly lower for other languages. This performance disparity can be attributed to three main factors: (1) *Benchmark difficulty*: Multi-SWE-bench is inherently more challenging than SWE-Bench Verified, with a higher proportion of medium and hard issues (77.1% for Multi-SWE-bench compared to 61.2% for SWE-Bench Verified, as calculated from Tab. 3). (2) *Method optimization bias*: The three methods are initially optimized for Python, resulting in a bias that limits their effectiveness across other languages. (3) *Language complexity*: Languages like TS and JS have asynchronous execution and varied runtimes, while C and C++ involve manual memory management and complex type systems, increasing the difficulty for issue resolving.

**High sensitivity to issue difficulty.** As shown in Tab. 5, LLM-based agents exhibit a performance that closely aligns with human-labeled difficulty, with resolved rates significantly decreasing as the issue difficulty increases from easy to hard. Among the evaluated models, Gemini-2.5-Pro achieves the highest resolved rates on both MagentLess and MopenHands across all difficulty levels, while Claude-3.7-Sonnet demonstrates the best performance on MSWE-agent. These results highlight the relative robustness of these models in handling issues of varying complexity compared to other LLMs. For hard-level issues, existing LLMs and agents are mostly ineffective, with resolved rates approaching zero. This phenomenon indicates the limitations of these LLMs and agents: they are primarily capable of addressing issues that human developers can resolve in under 15 minutes and are insufficient for handling more complex tasks.

Table 4: Resolved rate (%) of different models on Multi-SWE-bench.

| Methods | Models | All | Python | Java | TS | JS | Go | Rust | C | C++ |
|---|---|---|---|---|---|---|---|---|---|---|
| MagentLess | GPT-4o | 11.40 | 36.20 | 11.72 | 2.23 | 1.40 | 2.80 | 5.86 | 1.56 | 6.98 |
| | OpenAI-o1 | 16.23 | 48.20 | 21.09 | 5.80 | 5.06 | 4.44 | 7.11 | 1.56 | 5.43 |
| | OpenAI-o3-mini-high | 13.65 | 46.40 | 5.47 | 0.45 | 2.81 | 3.97 | 7.95 | 3.91 | 1.55 |
| | Claude-3.5-Sonnet | 13.56 | 42.40 | 14.84 | 4.91 | 1.97 | 5.14 | 5.02 | 1.56 | 3.88 |
| | Claude-3.7-Sonnet | 14.35 | 44.60 | 14.06 | 3.57 | 1.97 | 5.84 | 5.44 | 2.34 | 3.10 |
| | DeepSeek-V3 | 13.23 | 41.00 | 7.03 | 6.70 | 3.37 | 5.37 | 5.02 | 3.13 | 1.55 |
| | DeepSeek-R1 | 14.40 | 42.20 | 22.66 | 6.25 | 4.49 | 3.74 | 6.69 | 0.78 | 3.10 |
| | Qwen2.5-72B-Instruct | 8.26 | 26.80 | 10.94 | 4.46 | 0.84 | 1.40 | 2.51 | 0.78 | 0.78 |
| | Doubao-1.5-pro | 7.83 | 26.20 | 5.47 | 2.23 | 1.12 | 2.10 | 4.18 | 0.00 | 0.00 |
| | Doubao-1.5-thinking | 15.24 | 44.80 | 13.28 | 7.59 | 5.62 | 4.44 | 7.11 | 4.69 | 3.88 |
| | Gemini-2.5-Pro | 18.01 | 49.00 | 21.88 | 11.61 | 8.71 | 6.07 | 5.44 | 9.38 | 2.33 |
| | Llama-4-Maverick | 13.56 | 37.80 | 14.84 | 9.38 | 5.06 | 3.50 | 6.28 | 3.91 | 5.43 |
| MSWE-agent | GPT-4o | 6.29 | 18.80 | 12.50 | 0.45 | 0.84 | 2.34 | 2.09 | 1.56 | 2.33 |
| | OpenAI-o1 | 11.07 | 28.80 | 21.88 | 4.02 | 4.21 | 4.67 | 4.18 | 3.91 | 3.88 |
| | OpenAI-o3-mini-high | 10.74 | 28.60 | 16.41 | 4.91 | 4.21 | 3.97 | 5.02 | 2.34 | 5.43 |
| | Claude-3.5-Sonnet | 11.21 | 24.80 | 20.31 | 8.04 | 4.21 | 5.84 | 6.69 | 4.69 | 6.98 |
| | Claude-3.7-Sonnet | 17.17 | 45.80 | 23.44 | 11.16 | 4.78 | 5.37 | 6.69 | 8.59 | 11.63 |
| | DeepSeek-V3 | 4.55 | 4.20 | 11.72 | 2.68 | 2.53 | 4.44 | 5.86 | 2.34 | 7.75 |
| | DeepSeek-R1 | 2.95 | 2.00 | 9.38 | 5.80 | 1.40 | 2.10 | 2.09 | 0.78 | 6.20 |
| | Qwen2.5-72B-Instruct | 2.49 | 8.60 | 2.34 | 0.00 | 0.56 | 0.47 | 0.42 | 1.56 | 0.00 |
| | Doubao-1.5-pro | 4.88 | 12.40 | 7.03 | 1.79 | 1.40 | 2.10 | 1.67 | 2.34 | 6.20 |
| | Doubao-1.5-thinking | 10.46 | 30.60 | 11.72 | 7.14 | 1.69 | 4.21 | 3.35 | 0.78 | 4.65 |
| | Gemini-2.5-Pro | 14.63 | 27.80 | 28.91 | 8.93 | 7.58 | 9.81 | 10.04 | 9.38 | 8.53 |
| | Llama-4-Maverick | 3.52 | 2.00 | 15.63 | 4.46 | 2.25 | 2.80 | 2.51 | 0.00 | 6.98 |
| MopenHands | GPT-4o | 8.21 | 25.60 | 9.38 | 0.00 | 1.97 | 3.50 | 3.35 | 0.00 | 3.88 |
| | OpenAI-o1 | 6.10 | 16.00 | 3.91 | 0.45 | 3.65 | 3.74 | 2.51 | 3.13 | 3.88 |
| | OpenAI-o3-mini-high | 7.55 | 20.40 | 10.16 | 0.45 | 3.37 | 2.34 | 5.02 | 1.56 | 6.98 |
| | Claude-3.5-Sonnet | 15.24 | 39.00 | 14.84 | 11.61 | 1.97 | 6.78 | 12.13 | 3.13 | 12.40 |
| | Claude-3.7-Sonnet | 19.32 | 52.20 | 21.88 | 2.23 | 5.06 | 7.48 | 15.90 | 8.59 | 14.73 |
| | DeepSeek-V3 | 8.72 | 27.80 | 9.38 | 1.34 | 1.12 | 0.70 | 4.60 | 3.13 | 7.75 |
| | DeepSeek-R1 | 8.02 | 26.00 | 8.59 | 0.45 | 2.53 | 0.00 | 4.60 | 2.34 | 4.65 |
| | Qwen2.5-72B-Instruct | 2.02 | 4.40 | 3.13 | 0.00 | 0.84 | 1.40 | 1.67 | 0.78 | 2.33 |
| | Doubao-1.5-pro | 2.91 | 8.80 | 0.78 | 0.00 | 1.12 | 1.64 | 0.84 | 0.00 | 3.10 |
| | Doubao-1.5-thinking | 11.49 | 27.80 | 10.94 | 5.36 | 9.55 | 6.07 | 3.35 | 3.91 | 5.43 |
| | Gemini-2.5-Pro | 21.62 | 45.80 | 12.50 | 22.32 | 16.29 | 12.60 | 14.64 | 5.47 | 9.30 |
| | Llama-4-Maverick | 7.46 | 14.40 | 6.25 | 8.04 | 4.78 | 5.61 | 5.02 | 3.13 | 3.10 |

Table 5: Resolved rate (%) of different models on Multi-SWE-bench with varied difficulties.

| Models | MagentLess | | | MSWE-agent | | | MopenHands | | |
|---|---|---|---|---|---|---|---|---|---|
| | Easy | Medium | Hard | Easy | Medium | Hard | Easy | Medium | Hard |
| GPT-4o | 25.18 | 10.32 | 0.92 | 12.15 | 6.7 | 0.61 | 17.96 | 7.24 | 1.07 |
| OpenAI-o1 | 31.69 | 16.68 | 2.14 | 20.6 | 11.53 | 2.14 | 10.56 | 7.03 | 0.92 |
| OpenAI-o3-mini-high | 29.75 | 12.62 | 1.07 | 22.54 | 9.55 | 2.14 | 16.55 | 6.26 | 1.53 |
| Claude-3.5-Sonnet | 29.4 | 12.4 | 1.38 | 20.42 | 11.96 | 2.14 | 28.35 | 16.14 | 2.6 |
| Claude-3.7-Sonnet | 30.46 | 13.5 | 1.53 | 32.57 | 17.67 | 3.06 | 35.21 | 20.64 | 3.68 |
| DeepSeek-V3 | 27.11 | 12.84 | 1.68 | 8.8 | 4.17 | 1.38 | 17.78 | 8.45 | 1.23 |
| DeepSeek-R1 | 28.52 | 14.71 | 1.68 | 6.16 | 2.74 | 0.46 | 17.78 | 7.14 | 0.77 |
| Qwen2.5-72B-Instruct | 19.89 | 6.7 | 0.31 | 6.16 | 1.87 | 0.15 | 4.23 | 1.76 | 0.46 |
| Doubao-1.5-pro | 16.55 | 7.24 | 1.07 | 9.68 | 4.72 | 0.92 | 7.57 | 1.65 | 0.61 |
| Doubao-1.5-thinking | 30.99 | 14.71 | 2.3 | 20.6 | 10.54 | 1.53 | 24.30 | 9.99 | 2.45 |
| Gemini-2.5-Pro | 34.51 | 18.11 | 3.52 | 27.11 | 15.59 | 2.45 | 39.08 | 22.28 | 5.51 |
| Llama-4-Maverick | 30.46 | 12.07 | 0.92 | 6.87 | 3.18 | 1.07 | 16.20 | 5.82 | 2.14 |

## 6.2 Influencing Factors of Performance

**Varied resolved rate across different issue types.** Tab. 6 lists the performance of the three methods on Multi-SWE-bench across different issue types. Through a meticulous manual analysis of the annotation results in Sec. 3.5, we categorized all instances in Multi-SWE-bench into three issue types: bug fix, new feature, and feature optimization. We observe

Table 6: Resolved rate(%) on Multi-SWE-bench across issue types (Claude-3.7-Sonnet). BG refers to bug fixes, NF to new feature requests, and FO to feature optimizations.

| | MagentLess | | | MSWE-agent | | | MopenHands | | |
|---|---|---|---|---|---|---|---|---|---|
| | BF | NF | FO | BF | NF | FO | BF | NF | FO |
| Java | 10.94 | 2.34 | 0.78 | 17.97 | 3.91 | 1.56 | 17.97 | 3.12 | 0.78 |
| TS | 2.68 | 0.45 | 0.45 | 9.38 | 1.34 | 0.45 | 1.79 | 0.00 | 0.45 |
| JS | 1.97 | 0.00 | 0.00 | 4.21 | 0.56 | 0.00 | 3.65 | 1.12 | 0.28 |
| Go | 3.74 | 0.93 | 1.17 | 3.27 | 0.70 | 1.40 | 4.44 | 2.10 | 0.93 |
| Rust | 4.60 | 0.42 | 0.42 | 5.44 | 1.26 | 0.00 | 12.97 | 2.93 | 0.00 |
| C | 6.25 | 0.00 | 0.00 | 7.81 | 0.78 | 0.00 | 7.81 | 0.78 | 0.00 |
| C++ | 2.33 | 0.78 | 0.00 | 7.75 | 3.1 | 0.78 | 10.85 | 3.10 | 0.78 |

a consistent performance hierarchy across all methods and languages: bug fix issues are resolved with the highest success rates, followed by new features, with feature optimization being the most challenging. For instance, MSWE-agent achieves 17.97% on Java bug fixes but drops to 3.91% and 1.56% for new features and optimizations, respectively. MagentLess and MopenHands show a similar trend in all languages. These results highlight a fundamental limitation of current agent-based methods: they are more effective at localized, symptom-driven repairs, but struggle with semantically demanding tasks such as implementing new functionality or refining existing behavior.

**Performance drops as fix patch length increases.** As shown in Fig. 2, the length of fix patches significantly impacts the resolved rate, with shorter patches generally leading to higher success rates. Specifically, in the majority of cases, issues with descriptions >600 tokens exhibit a resolved rate approximately 50% lower than that of issues with descriptions <200 tokens. For all three methods, the resolved rate for very long fix patches (>1000 tokens) drops sharply for very long fix patches (over 1,000 tokens), with a resolved rate approaching zero. This indicates that long patches, requiring broader code modifications, pose greater challenges, especially for methods not optimized for complex tasks.

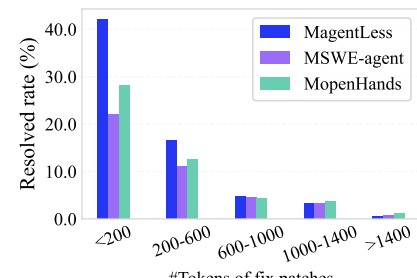

Figure 2: Influence of fix patch length on resolved rate (%) across methods.

**Cross-file fix patches lead to reduced effectiveness.** Fig. 3 illustrates the relationship between the number of files modified by fix patches and the resolved rate. Consistent with the observation in Fig. 12, resolved rate drops significantly as the number of modified files increases across all three methods. This trend highlights the potential challenge of understanding and resolving issues that require changes across multiple files, which may demand more intricate handling or coordination between different parts of the repository. For issues resolved by modifications in a single file, MagentLess outperforms MSWE-agent and MopenHands, which suggests that MagentLess is more effective at resolving issues within the scope of a single file.

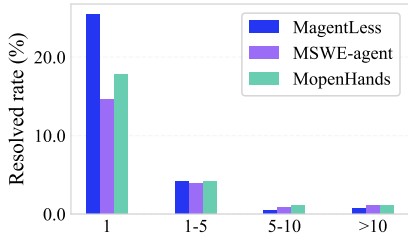

Figure 3: Influence of number of files modified by fix patches across methods.

**Effect of Potential Data Leakage.** Since Multi-SWE-bench is constructed from historical GitHub issues, there exists a possibility that these issues are included within the pre-training datasets of the LLMs we employ. Here, we examine the impact of potential data leakage on Claude-3.5-Sonnet, which has a clearly stated knowledge cutoff date (i.e., 2024-04-01). Tab. 7 shows no significant difference in performance before and after this cutoff. For MSWE-agent, the resolved rate after the knowledge cutoff is even higher than before on easy tasks. Furthermore, the data leakage degree is identical for methods within the same models. This uniformity ensures a fair basis for comparing different methods in Tab. 4.

Table 7: Resolved rate (%) before/after knowledge cutoff on Claude-3.5-Sonnet.

|  | MagentLess | | MSWE-agent | | MopenHands | |
|---|---|---|---|---|---|---|
|  | **Before** | **After** | **Before** | **After** | **Before** | **After** |
| Easy | 8.32 | 7.94 | 10.10 | 15.87 | 11.09 | 15.87 |
| Medium | 2.49 | 3.74 | 4.10 | 8.41 | 5.97 | 4.67 |
| Hard | 1.09 | 0.51 | 1.53 | 2.55 | 1.75 | 1.53 |
| All | 3.79 | 2.73 | 5.15 | 6.56 | 6.34 | 4.92 |

We also conducted large-scale analysis experiments, with results provided in Appendix E.

### 6.3 Cost

Tab. 8 presents the average cost ($) per issue on Multi-SWE-bench. Notably, DeepSeek-V3, DeepSeek-R1, Llama-4-Maverick, and Qwen2.5-72B-Instruct achieve the lowest cost per resolved issue, staying below $0.03, benefiting from their cost-efficient pricing. In contrast, OpenAI-o1 is the most expensive model, due to its high token price ($15 per million input tokens). Overall, MagentLess tends to result in higher costs than MSWE-agent, as it follows a fixed

Table 8: Average cost ($) per issue of different models and methods on Multi-SWE-bench.

| Models | MagentLess | MSWE-agent | MopenHands |
|---|---|---|---|
| GPT-4o | 0.2021 | 0.1919 | 0.0943 |
| OpenAI-o1 | 1.2549 | 1.0747 | 0.4631 |
| OpenAI-o3-mini-high | 0.1154 | 0.0738 | 0.0474 |
| Claude-3.5-Sonnet | 0.2588 | 0.1470 | 0.2142 |
| Claude-3.7-Sonnet | 0.2966 | 0.1760 | 0.2127 |
| DeepSeek-V3 | 0.0094 | 0.0080 | 0.0065 |
| DeepSeek-R1 | 0.0170 | 0.0071 | 0.0148 |
| Qwen2.5-72B-Instruct | 0.0115 | 0.0105 | 0.0083 |
| Doubao-1.5-pro | 0.0134 | 0.0049 | 0.0034 |
| Doubao-1.5-thinking | 0.0557 | 0.0287 | 0.0247 |
| Gemini-2.5-Pro | 0.1538 | 0.0990 | 0.1689 |
| Llama-4-Maverick | 0.0214 | 0.0081 | 0.0085 |

workflow regardless of task difficulty. In comparison, the workflows in MSWE-agent and Mopen-Hands are dynamically controlled by LLMs, allowing more flexible interaction turns. For simpler tasks, they typically require fewer interactions, resulting in lower overall costs.

# 7   Conclusions and Future Works

We introduce Multi-SWE-bench, a multilingual benchmark for issue resolving, consisting of $2,132$ human-validated GitHub instances on 8 widely used programming languages. Based on this benchmark, we evaluate 12 popular models using three representative methods and conduct a thorough analysis of the results. Looking ahead, we plan to scale Multi-SWE-bench to more instances, languages, and modalities. Beyond issue resolving, we would like to incorporate a broader range of software engineering tasks into our benchmark such as end-to-end project generation [46, 33], runtime environment setup [43, 15, 11], bug reproduction [36, 35] and localization [14], and software testing and maintenance [22, 29].

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

# A Manual Verification Results

In Tab. 9 and Fig. 4, we present the statistics from the manual verification results. As shown in Tab. 9, the majority of instances show no significant issues and receive high scores, which confirms the overall quality of the repositories selected in Section 3.1. As part of the manual annotation process in Multi-SWE-bench, we recorded the estimated time required to resolve each issue, categorized into four buckets: ≤15 minutes, 15 minutes–1 hour, 1–4 hours, and ≥4 hours (Fig. 4). Unlike SWE-Bench, we use this time-based annotation to define difficulty levels across all languages: easy (≤15 mins), medium (15 mins–1h), and hard (≥1h). From Figure 4,

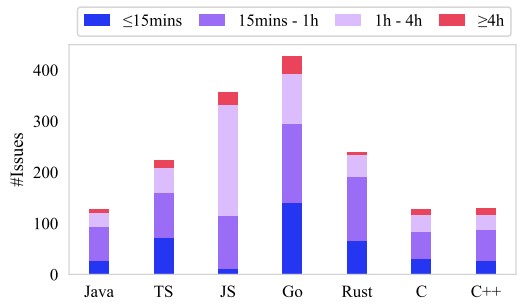

Figure 4: Distribution of estimated time consumption of issues in Multi-SWE-bench.

we can see that JavaScript (JS) emerges as the most difficult language, as it generally requires more time for developers to resolve the issues. In contrast, TypeScript (TS) and Rust appear to be easier, with most issues being resolved by humans within an hour.

Table 9: Scoring statistics for Multi-SWE-bench from the verification questionnaire.

| Languages | Q2.1 Serious Issue Flag | | Q3.1 Clarity of Issue Description | | | | Q4.1 Coverage of Unit Tests | | | |
|---|---|---|---|---|---|---|---|---|---|---|
| | #Score 0 | #Score 1 | #Score 0 | #Score 1 | #Score 2 | #Score 3 | #Score 0 | #Score 1 | #Score 2 | #Score 3 |
| Java | 146 | 10 | 2 | 2 | 44 | 98 | 10 | 5 | 17 | 114 |
| TypeScript | 382 | 8 | 5 | 56 | 121 | 200 | 31 | 76 | 133 | 142 |
| JavaScript | 586 | 4 | 0 | 6 | 13 | 567 | 55 | 172 | 305 | 54 |
| Go | 579 | 26 | 5 | 10 | 276 | 288 | 44 | 100 | 151 | 284 |
| Rust | 328 | 11 | 4 | 20 | 165 | 139 | 23 | 50 | 74 | 181 |
| C | 200 | 6 | 2 | 4 | 115 | 79 | 13 | 55 | 83 | 49 |
| C++ | 162 | 7 | 0 | 6 | 96 | 60 | 7 | 21 | 45 | 89 |

# B Multi-SWE-RL

**Community Introduction.** Multi-SWE-RL is an open-source community aimed at developing high-quality RL training datasets for complex software engineering tasks. Its purpose is to serve as the foundational infrastructure for training fully autonomous agents capable of addressing real-world software engineering challenges, paving the way toward achieving AGI. The need for such a community has become increasingly urgent as the potential of RL continues to expand. Notable models such as DeepSeek-R1 [12], OpenAI o1 [17], and o3 [27] have demonstrated the power of RL, even with simple, rule-based reward signals. In light of these advancements, we are firmly convinced that "*scaling RL in real-world environments is the path toward human-like intelligence*". However, the creation of such interactive environments and data trajectories is extremely challenging. For instance, the development of our Multi-SWE-bench took about one year to produce just high-quality 2,132 instances. Therefore, we launched the Multi-SWE-RL community to harness the power of open-source collaborative contributions for building diverse RL environments.

**Community Initialization.** To bootstrap the Multi-SWE-RL community, we release an initial dataset comprising 4,723 issue-resolving instances spanning 76 widely-used open-source repositories and 7 programming languages: Java, TypeScript, JavaScript, Go, Rust, C, and C++. Each instance is equipped with a fully containerized execution environment to ensure reproducibility and ease of integration. This dataset was constructed using the same pipeline as Multi-SWE-bench, excluding the manual verification process described in Sec. 3.5. Details about this release are available at Hugging Face dataset and Multi-SWE-RL contribution board. We envision this initial release as a spark—igniting broader community collaboration and fueling the construction of scalable, high-quality RL environments for real-world software engineering.

**Contribution Guidelines and Recognition.** We welcome contributions from the community to expand the Multi-SWE-bench and Multi-SWE-RL. To help new contributors get started, we provide a detailed demo that walks through the process of creating an issue-resolving instance, available at Contribution-demo.md. To recognize and incentivize community contributions, we maintain a rolling

update schedule through periodic arXiv updates or follow-up technical reports, with new versions released every three months. Each update may include:

- Newly added benchmarks for additional programming languages in Multi-SWE-bench, with new authors and contributors;

- Newly contributed data to Multi-SWE-RL, with new authors and contributors;

- Newly reported performance results from RL trials on Multi-SWE-bench using Multi-SWE-RL data, with new authors and contributors;

- Newly open-sourced RL models with significantly enhanced performance, with new authors and contributors.

Our contribution incentive policy is detailed at Incentive-plan.md. We are committed to continuously refining our contribution strategy to encourage sustained open-source engagement, and we warmly invite the community to take part in shaping and scaling this collaborative effort.

## C  Troubleshooting

During the construction of Multi-SWE-bench and Multi-SWE-RL, we encountered a range of practical and non-obvious challenges. We document the key issues below to facilitate reproducibility and guide future community contributions:

- Test log inconsistency. The number of test cases differs between Test.log and Fix.log, as fix.patch may optimize control flow, eliminate redundant coverage, or merge test paths, which is commonly observed in repositories such as preactjs/preact.

- Pre-fix build failures. Certain repositories fail to compile or execute tests before applying fix.patch, due to newly introduced symbols (e.g., functions or variables) in test.patch that are undefined without the fix.

- Binary artifacts in C&C++. Agent runs may generate compiled binaries (e.g., ".o", ".bin") that block "git apply". We currently strip these via hard-coded filtering, though more robust handling is needed.

- Evaluation nondeterminism. Java and C tests occasionally exhibit unstable behavior due to excessive thread concurrency, leading to inconsistent run.log outcomes. We mitigate this by reducing parallelism during evaluation.

- Name casing mismatches. Some test names appear in lowercase in test.log but in uppercase in fix.log. We normalize all test names to lowercase to ensure alignment.

- Unstable test identifiers. Some test names are dynamically generated with timestamps or random suffixes, making them non-deterministic. Such instances are excluded.

- Log interleaving in Java. In some Java projects, test outputs from concurrent threads are interleaved without delimiters, making rule-based log parsing infeasible. This is likely due to unsynchronized multi-threaded logging.

## D  Implementations of Issue Resolving Methods

To support the multilingual environment of Multi-SWE-bench we adapt the issue resolving methods Agentless [39], SWE-agent [40], and OpenHands [37]. The details of their adaptation and implementation are outlined as follows:

- **Agentless**[4]→**MagentLess**[5]: Agentless addresses the issue resolving task through a multi-stage fixed workflow, including hierarchical fault localization, code repair, and candidate patch selection via regression and reproduction tests. In MagentLess, we made the following key modifications to support multilingual adaptation and improve scalability:

    1. We revised all prompts to accommodate the newly added languages.

---

[4]https://github.com/OpenAutoCoder/Agentless
[5]https://github.com/multi-swe-bench/MagentLess

2. We replaced all file skeleton inputs with full file content, as extracting file skeletons is challenging in some programming languages.

3. We implemented function and class extraction for all languages using Tree-sitter[6].

4. We pruned the extracted repository structures by retaining only files and directories with specific extensions, as repositories in certain languages (e.g., TypeScript) often contain an excessive number of files that may exceed LLM context limits.

5. We removed the candidate patch selection stage and retained only fault localization and code repair, as regression and reproduction testing is cumbersome to implement across languages and falls outside the scope of this work.

- **SWE-agent**[7]→**MSWE-agent**[8]: SWE-agent is an agent-based approach that solves issues through multi-turn interactions via a predefined agent-computer interface (ACI). To support Multi-SWE-bench, we developed MSWE-agent with the following modifications:

  1. We revised all prompts to accommodate the newly added languages.

  2. We truncated overly long environment observations to ensure stable agent execution.

  3. We added ".gitignore" to exclude compiled artifacts (e.g., ".o", ".bin") in languages like C/C++, which could otherwise interfere with "git apply".

  4. We fixed language-specific commands that caused crashes or non-terminating behavior during execution to ensure stable agent execution.

- **OpenHands**[9]→**MopenHands**[10]: OpenHands is a widely adopted platform for building software development agents. In MopenHands, we made the following key modifications to support multilingual adaptation:

  1. We revised all prompts to support the newly added programming languages.

  2. We added ".gitignore" to exclude compiled artifacts, as also done in MSWE-agent.

  3. We fixed several implementation bugs, including an issue where "CmdRunAction" incorrectly rendered tab characters (\t) as spaces in "git diff" outputs, making patches unapplicable. To resolve this, we redirected the diff to a file and read it using "FileReadAction", which proved especially important in languages like Go.

Despite our efforts to adapt these methods, there still remains substantial room for improvement, particularly in language-specific adaptation and overall robustness. We welcome community collaboration to further advance their capabilities.

For all LLM-based tasks in the issue resolving methods, we used a temperature setting of $0.8$ and employed top-k sampling to ensure a balance between creativity and consistency in the generated outputs. These hyperparameters were kept consistent across all methods and languages to ensure comparability with other models. This uniform setting allows for a fair evaluation of the performance of different methods in the multilingual environment of Multi-SWE-bench.

# E   Additional Experimental Results

## E.1   Multi-SWE-bench Mini

To facilitate faster and more cost-effective evaluations for both the research community and industry, we have created a mini version of Multi-SWE-bench, called Multi-SWE-bench Mini. This smaller subset contains 400 instances in total. For the construction of this mini version, we randomly selected 50 instances restricted to the same difficulty distribution for each language from each of the eight languages: Python, Java, TypeScript, JavaScript, Go, Rust, C, and C++, while ensuring that the difficulty distribution remained consistent across all languages. The experimental results of Multi-SWE-bench Mini is shown in Tab. 10.

Table 10: Resolved rate (%) of different models on Multi-SWE-bench Mini.

| Models | MagentLess | | | | MSWE-agent | | | | MopenHands | | | |
|---|---|---|---|---|---|---|---|---|---|---|---|---|
| | All | Easy | Medium | Hard | All | Easy | Medium | Hard | All | Easy | Medium | Hard |
| GPT-4o | 7.50 | 16.00 | 7.47 | 0.79 | 5.25 | 9.00 | 6.32 | 0.79 | 6.75 | 14.00 | 7.47 | 0.00 |
| OpenAI-o1 | 10.75 | 16.00 | 14.37 | 1.59 | 7.25 | 14.00 | 8.05 | 0.79 | 4.75 | 7.00 | 6.90 | 0.00 |
| OpenAI-o3-mini-high | 9.00 | 19.00 | 9.77 | 0.00 | 7.50 | 16.00 | 7.47 | 0.79 | 5.75 | 12.00 | 5.75 | 0.79 |
| Claude-3.5-Sonnet | 8.25 | 18.00 | 7.47 | 1.59 | 9.25 | 16.00 | 11.49 | 0.79 | 12.25 | 22.00 | 14.37 | 1.59 |
| Claude-3.7-Sonnet | 9.25 | 19.00 | 9.77 | 0.79 | 15.25 | 28.00 | 18.39 | 0.79 | 18.25 | 33.00 | 20.69 | 3.17 |
| DeepSeek-V3 | 8.00 | 17.00 | 8.05 | 0.79 | 5.00 | 8.00 | 6.32 | 0.79 | 7.50 | 16.00 | 8.05 | 0.00 |
| DeepSeek-R1 | 10.00 | 18.00 | 12.07 | 0.79 | 3.75 | 7.00 | 4.60 | 0.00 | 7.00 | 14.00 | 7.47 | 0.79 |
| Qwen2.5-72B-Instruct | 6.00 | 16.00 | 4.60 | 0.00 | 1.00 | 1.00 | 1.72 | 0.00 | 2.25 | 3.00 | 2.87 | 0.79 |
| Doubao-1.5-pro | 4.50 | 9.00 | 4.60 | 0.79 | 3.75 | 5.00 | 5.17 | 0.79 | 2.00 | 2.00 | 3.45 | 0.00 |
| Doubao-1.5-thinking | 9.50 | 19.00 | 10.92 | 0.00 | 7.75 | 13.00 | 10.34 | 0.00 | 8.75 | 19.00 | 9.20 | 0.00 |
| Gemini-2.5-Pro | 12.25 | 25.00 | 12.64 | 1.59 | 9.75 | 15.00 | 13.79 | 0.00 | 18.00 | 27.00 | 21.84 | 5.56 |
| Llama-4-Maverick | 10.25 | 21.00 | 11.49 | 0.00 | 4.25 | 9.00 | 4.02 | 0.79 | 5.50 | 7.00 | 7.47 | 1.59 |

## E.2  Performance across Various Methods

In this subsection, we evaluate the methods' performance from two aspects: (1) their ability to locate issues and generate fix patches, and (2) for agent-based methods, i.e., MSWE-agent and MopenHands, their efficiency in terms of the number of interaction turns required to resolve the issues.

**Prioritizing accurate locating over editing and reproducing.** MagentLess, MSWE-agent, and MopenHands generally resolve issues through two key steps: issue location and code editing to resolve the issue. To provide a more detailed analysis of how existing LLMs and methods perform across these steps, we present the issue flow in Fig. 5. An issue is considered successfully located if the fix patches generated by the LLMs cover all the files of ground truth fix patches. As shown in Fig. 5, all three methods generally fail to locate issues more often than they succeed. Accurate issue localization is fundamental to the overall success of the resolution process, serving as a prerequisite for effective code editing. Compared to MopenHands, MagentLess achieves more accurate issue localization but struggles more with the code editing step, leading to a lower overall resolved rate. This disparity is particularly evident on Claude-3.7-Sonnet. This underscores the need for a balanced method that not only prioritizes precise issue identification but also enhances the model's ability to generate effective fixes.

**Number of turns required by MSWE-agent and MopenHands.** Both MSWE-agent and Mopen-Hands resolve the issue by multi-turn interactions. Fig. 6 shows the distribution of turns for successfully resolved an issue. The absence of a corresponding box plot indicates cases where no issues were successfully resolved, such as MSWE-agent with Qwen2.5-72B-Instruct on C++. The number of interaction turns required by two methods differs across models and languages. Specifically, MopenHands resolves issues in fewer turns than MSWE-agent when using GPT-4o for Java, whereas MSWE-agent requires fewer turns when resolving Python issues. However, MopenHands exhibits a rather higher degree of dispersion in the number of interaction turns compared to MSWE-agent, which is particularly evident on OpenAI-o3-mini-high. This suggests that MopenHands' performance is less stable across different issues, requiring a varying number of turns depending on the complexity or nature of the issue.

## E.3  Performance across Different Repositories

To understand how repository characteristics affect performance, we examine two factors: (1) repository quality, which includes the number of stars, forks, PRs, and issues, and (2) repository complexity, which includes the number of code lines and files, and the language entropy.

**Performance across repositories of varying quality.** To assess repository quality, we examine key metrics including the number of stars, forks, PRs, and issues. Fig. 7 illustrates the average resolved rate across LLMs for the three methods in relation to the number of stars and forks. Similarly, Fig. 8 shows the average resolved rate in relation to the number of issues and PRs. Both Fig. 7 and Fig. 8

---

[6]`https://tree-sitter.github.io`

[7]`https://github.com/SWE-agent/SWE-agent`

[8]`https://github.com/multi-swe-bench/MSWE-agent`

[9]`https://github.com/All-Hands-AI/OpenHands`

[10]`https://github.com/multi-swe-bench/MopenHands`

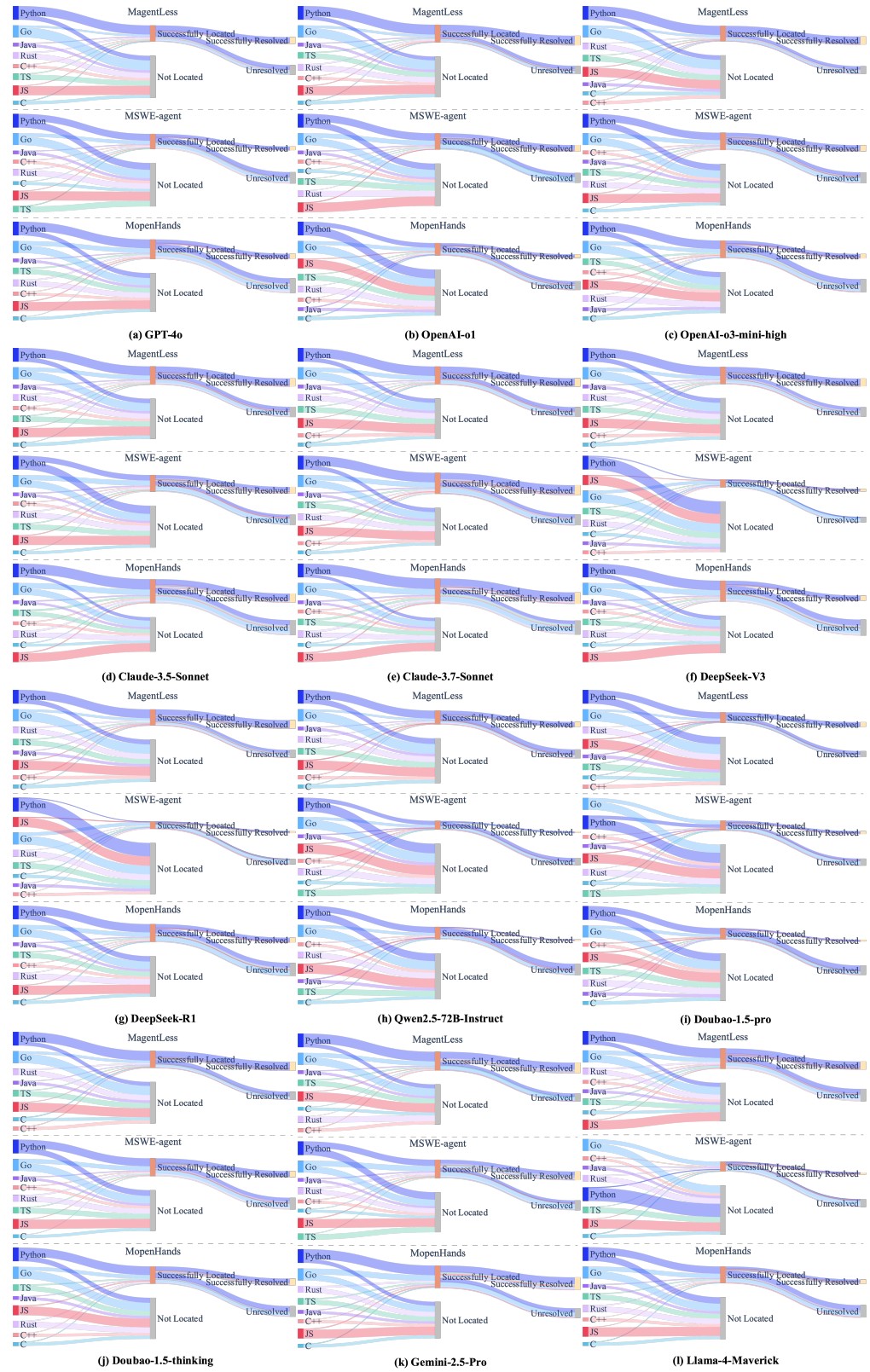

Figure 5: Issue flow from locating to resolving.

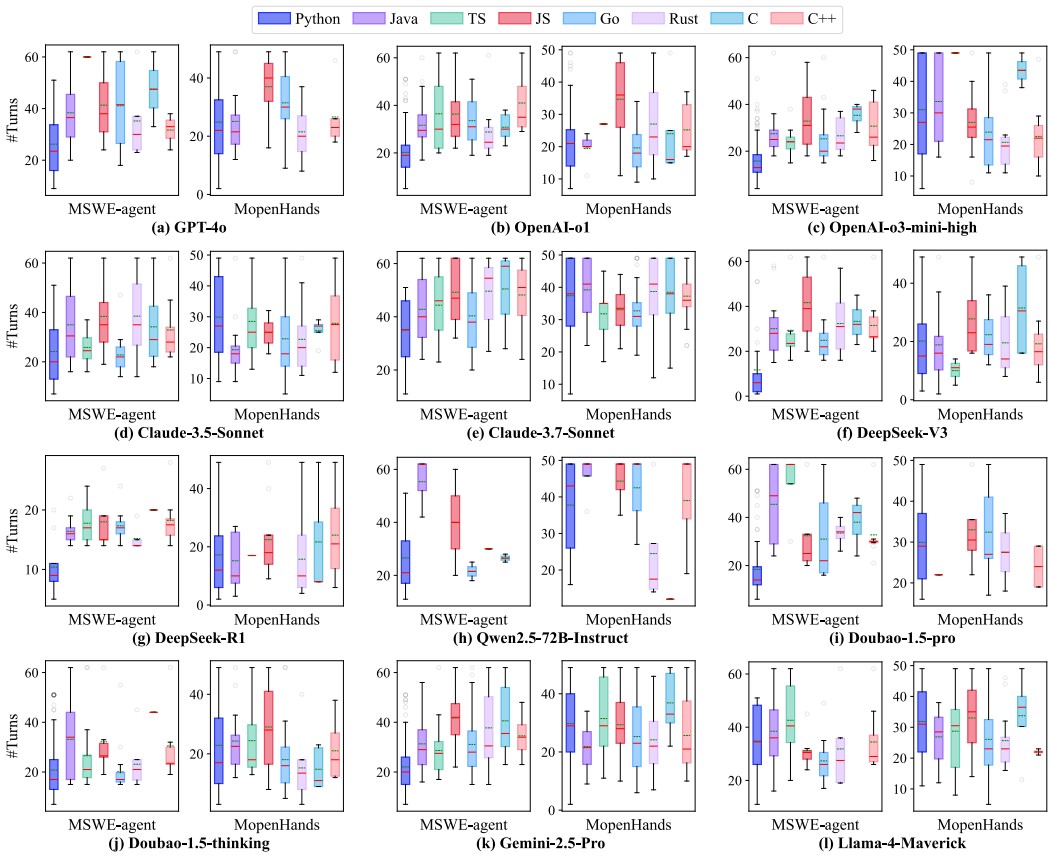

Figure 6: Number of turns required across different programming languages.

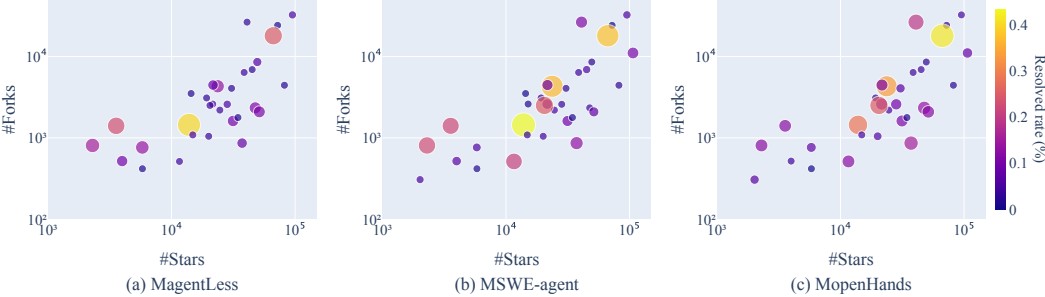

Figure 7: Relationship between resolved rate and the number of stars and forks of a repository.

exhibit a general positive correlation between #Stars and #Forks, as well as #Issues and #PRs across the majority of repositories. Furthermore, repositories with higher resolved rates tend to cluster in the upper-right quadrant of both Fig. 7 and Fig. 8, suggesting that repositories with greater activity and community engagement (i.e., higher counts of stars, forks, issues, and PRs) are typically associated with a higher resolved rate. This trend is particularly evident for the MSWE-agent and MopenHands. In contrast, MagentLess exhibits relatively low variation in resolved rates across both Fig. 7 and Fig. 8, underscoring an important observation: while a greater number of stars, forks, issues, and PRs tend to correlate with higher resolved rates, these metrics do not provide a guarantee of a repository's issue-resolving effectiveness.

**Performance across repositories with different levels of complexity.** To evaluate repository complexity, we consider several key metrics: the number of lines of code (#LoC), the number of files (#Files), and language entropy. Let $L = \{l_1, l_2, \cdots, l_n\}$ represent the set of programming languages

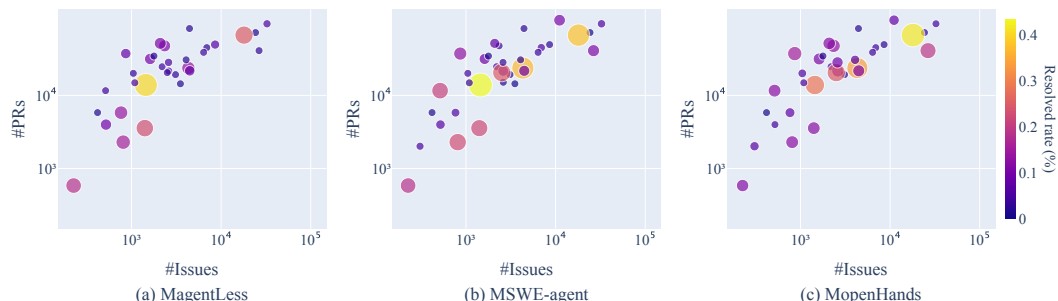

Figure 8: Relationship between resolved rate and the number of issues and PRs of a repository.

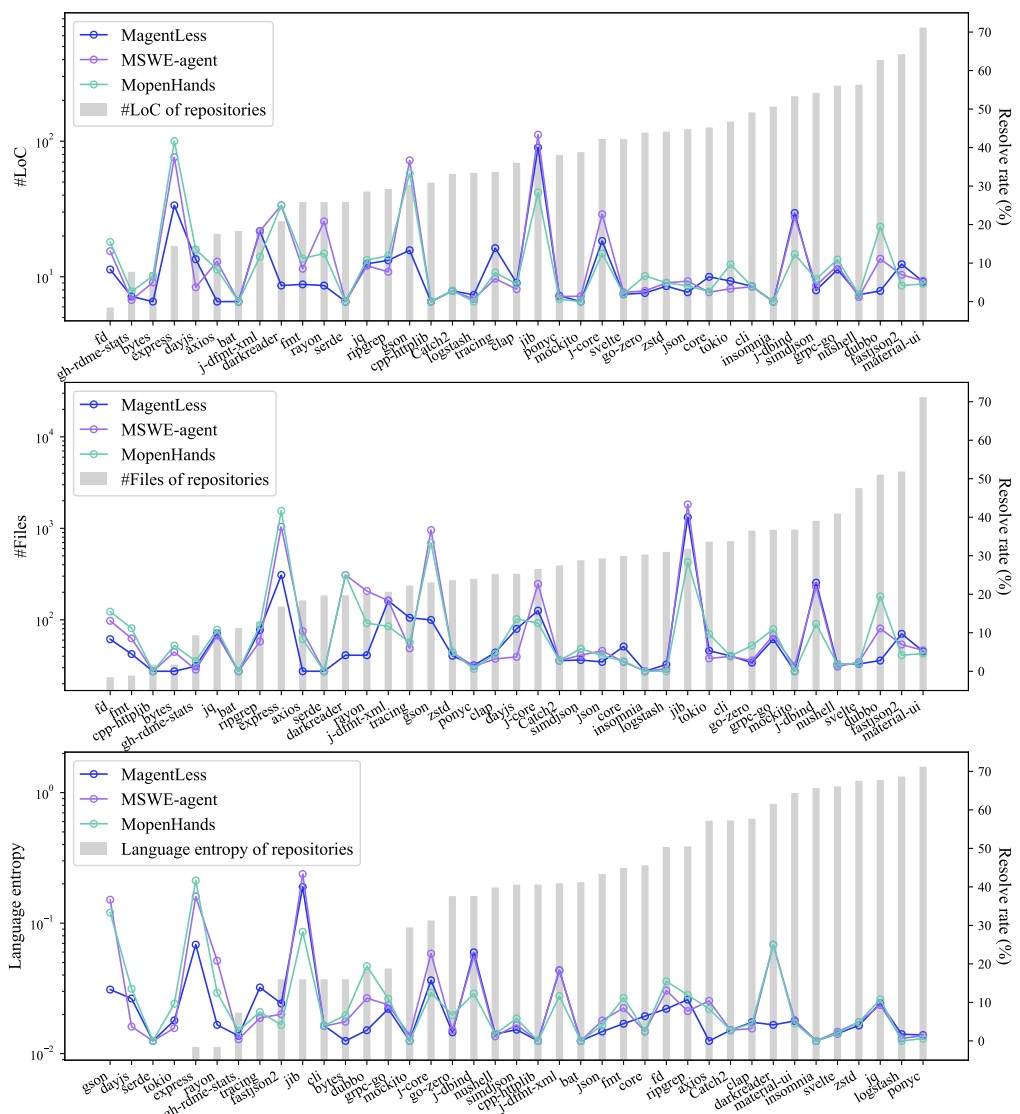

Figure 9: Relation between resolved rate and the repository complexity on Multi-SWE-bench.

used in the repository, with corresponding proportions $\{p_1, p_2, \cdots, p_n\}$. The language entropy of

the repository is then calculated as:

$$H(L) = - \sum_{i=1}^{n} p_i \log(p_i)$$

where $p_i$ denotes the proportion of the repository written in language $l_i$. The average resolved rate across nine base LLMs with different repository complexity is presented in Fig. 9.

Fig. 9 shows a consistent trend in the resolved rate across varied repository complexity: All three methods exhibit fluctuations in performance with changes in #LoC, #Files, and language entropy, generally decreasing as the repository complexity increases. For the impact of #LoC, as #LoC increases, the resolved rate tends to decrease. However, Java-based repositories, such as gson, jib, j-core, j-dbind, and dubbo, show higher resolved rates despite their larger size. This suggests that factors beyond code size, such as lower language entropy, modularity, well-documented code, and adherence to standardized practices, play a significant role in improving performance. For example, the gson repository demonstrates nearly-zero language entropy in Fig. 9. Similarly, the impact of #Files follows a trend similar to #LoC. The impact of language entropy shows a clearer trend than that of #LoC and #Files: repositories with lower entropy typically achieve higher resolved rates. This indicates that code simplicity and consistency play a crucial role in improving issue-resolving effectiveness on a repository.

### E.4 Detailed Results across various difficulty levels

Tab. 11 presents detailed results across various difficulty levels for each of the eight programming languages. From Tab. 11, it can be seen that Java, C, and C++ emerge as the most challenging languages, particularly at the hard difficulty level, where most models and methods fail to resolve even a single issue. This highlights the increased complexity these languages present in comparison to others. Furthermore, across all three difficulty levels, existing models and methods consistently perform better on Python than on the other languages. This suggests that these models and methods have an inherent bias towards Python. These findings underscore the importance of evaluation in a multilingual environment to fully study the capabilities and limitations of current LLMs and methods.

### E.5 Influence of Issue Description

In this subsection, we aim to examine the impact of issue description length on issue-resolving performance. Fig. 10 illustrates the distribution of issue lengths (in tokens) in Multi-SWE-bench, which follows a power law, with the majority of issues being under 1,000 tokens. To explore the effect of description length, the issues are categorized into 5 intervals: <100, 100-400, 400-700, 700-1000, and >1000 tokens, as shown in Fig. 11. The absence of a corresponding bars indicates cases where no issues are successfully resolved.

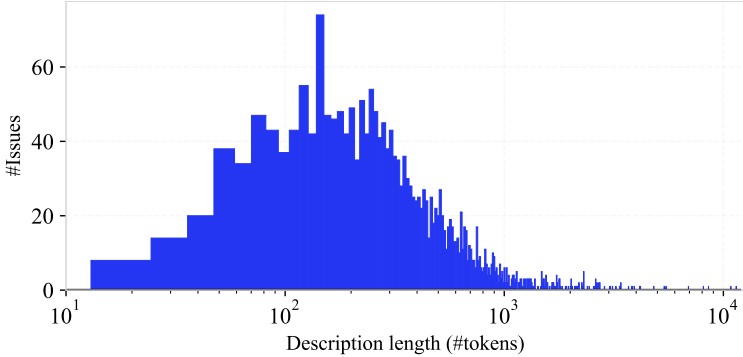

Figure 10: Histogram of issue description length (#tokens).

As shown in Fig. 11, there is no consistent relationship between issue description length and resolved rate. For example, in Python, issues with longer descriptions tend to have lower resolved rates, whereas in Go, longer descriptions are associated with higher rates. This discrepancy arises from two

Table 11: Resolved rate (%) of different models on Multi-SWE-bench across various difficulty levels.

| Models | Easy All | Python | Java | TS | JS | Go | Rust | C | C++ | Medium All | Python | Java | TS | JS | Go | Rust | C | C++ | Hard All | Python | Java | TS | JS | Go | Rust | C | C++ |
|---|---|---|---|---|---|---|---|---|---|---|---|---|---|---|---|---|---|---|---|---|---|---|---|---|---|---|---|
| **MagentLess** | | | | | | | | | | | | | | | | | | | | | | | | | | | |
| GPT-4o | 25.18 | 55.15 | 22.22 | 4.17 | 20.00 | 6.38 | 13.64 | 3.33 | 17.86 | 10.32 | 27.97 | 13.85 | 1.14 | 1.90 | 1.96 | 1.59 | 6.78 | 0.92 | 0.41 | 2.22 | 0.00 | 1.56 | 1.66 | 0.41 | 6.38 | 0.00 | 0.00 |
| OpenAI-o1 | 31.69 | 68.04 | 40.74 | 11.11 | 20.00 | 6.38 | 16.67 | 3.33 | 14.29 | 16.68 | 40.23 | 24.62 | 4.55 | 9.52 | 5.88 | 3.17 | 5.08 | 2.14 | 0.75 | 2.22 | 0.00 | 1.56 | 2.49 | 0.75 | 0.00 | 0.00 | 0.00 |
| OpenAI-o3-mini-high | 29.75 | 67.01 | 14.81 | 1.39 | 30.00 | 9.93 | 22.73 | 6.67 | 3.57 | 12.62 | 38.31 | 4.62 | 0.00 | 4.76 | 1.31 | 2.38 | 1.85 | 1.07 | 0.83 | 4.44 | 0.00 | 0.00 | 3.73 | 0.75 | 4.26 | 0.00 | 0.00 |
| Claude-3.5-Sonnet | 29.40 | 61.86 | 37.04 | 11.11 | 30.00 | 11.35 | 9.09 | 3.33 | 10.71 | 12.40 | 34.10 | 13.85 | 2.27 | 1.90 | 3.27 | 3.17 | 1.85 | 1.38 | 0.83 | 6.67 | 0.00 | 1.56 | 0.83 | 4.26 | 0.00 | 0.00 | 0.00 |
| Claude-3.7-Sonnet | 30.46 | 64.43 | 33.33 | 10.61 | 20.00 | 13.48 | 10.61 | 3.33 | 7.14 | 13.50 | 35.63 | 13.85 | 3.41 | 3.81 | 3.92 | 3.17 | 1.85 | 1.53 | 1.66 | 6.67 | 0.00 | 1.56 | 1.66 | 0.75 | 4.26 | 0.00 | 0.00 |
| DeepSeek-V3 | 27.11 | 57.73 | 18.52 | 5.56 | 30.00 | 9.93 | 6.67 | 6.67 | 7.14 | 12.84 | 34.87 | 6.15 | 3.41 | 4.76 | 3.92 | 0.79 | 1.85 | 1.68 | 0.41 | 4.44 | 0.00 | 1.56 | 1.49 | 0.00 | 4.26 | 0.00 | 0.00 |
| DeepSeek-R1 | 28.52 | 58.76 | 51.85 | 11.11 | 30.00 | 7.80 | 15.15 | 0.00 | 12.84 | 14.71 | 36.02 | 23.08 | 6.82 | 7.62 | 1.96 | 3.97 | 1.85 | 1.68 | 1.49 | 6.67 | 0.00 | 1.56 | 1.66 | 0.00 | 2.13 | 0.00 | 0.00 |
| Qwen2.5-72B-Instruct | 19.89 | 44.33 | 33.33 | 6.94 | 20.00 | 3.55 | 15.15 | 0.00 | 3.57 | 6.70 | 18.39 | 7.69 | 4.55 | 0.00 | 0.65 | 1.59 | 0.00 | 0.31 | 0.00 | 0.00 | 0.00 | 1.56 | 0.00 | 0.00 | 0.00 | 0.00 | 0.00 |
| Doubao-1.5-pro | 16.55 | 39.18 | 14.81 | 1.39 | 10.00 | 3.55 | 6.06 | 0.00 | 0.00 | 7.24 | 20.31 | 4.62 | 3.41 | 2.61 | 2.61 | 1.59 | 1.85 | 1.07 | 0.00 | 4.44 | 0.00 | 0.83 | 1.24 | 0.75 | 0.00 | 0.00 | 0.00 |
| Doubao-1.5-thinking | 30.99 | 62.37 | 33.33 | 16.67 | 30.00 | 9.22 | 10.61 | 0.00 | 7.14 | 14.71 | 37.93 | 7.69 | 9.52 | 0.00 | 0.65 | 3.17 | 3.70 | 1.68 | 2.13 | 8.89 | 0.00 | 1.56 | 2.13 | 0.75 | 0.00 | 2.27 | 0.00 |
| Gemini-2.5-Pro | 34.51 | 67.01 | 51.85 | 18.06 | 30.00 | 14.18 | 15.15 | 20.00 | 18.11 | 21.54 | 42.15 | 21.54 | 13.64 | 14.29 | 3.27 | 3.17 | 1.69 | 3.52 | 2.90 | 11.11 | 0.00 | 1.56 | 5.39 | 0.75 | 0.00 | 4.55 | 2.38 |
| Llama-4-Maverick | 30.46 | 61.34 | 40.74 | 19.44 | 30.00 | 7.80 | 12.12 | 10.00 | 14.29 | 12.07 | 26.44 | 12.31 | 7.95 | 10.48 | 2.61 | 4.76 | 3.70 | 0.92 | 1.66 | 2.22 | 0.00 | 0.00 | 1.66 | 0.00 | 2.13 | 0.00 | 0.00 |
| **MSWE-agent** | | | | | | | | | | | | | | | | | | | | | | | | | | | |
| GPT-4o | 12.15 | 25.77 | 22.22 | 0.00 | 0.00 | 5.67 | 1.52 | 6.67 | 7.14 | 6.70 | 16.09 | 15.38 | 1.14 | 0.95 | 1.31 | 3.17 | 0.00 | 1.69 | 0.83 | 2.22 | 0.00 | 0.00 | 0.83 | 0.75 | 2.13 | 0.00 | 0.00 |
| OpenAI-o1 | 20.60 | 40.72 | 48.15 | 4.17 | 10.00 | 8.51 | 6.06 | 10.00 | 7.14 | 11.53 | 24.14 | 23.08 | 4.55 | 6.67 | 3.92 | 3.97 | 3.70 | 2.14 | 2.49 | 8.89 | 0.00 | 3.13 | 2.49 | 0.75 | 2.13 | 0.00 | 0.00 |
| OpenAI-o3-mini-high | 22.54 | 42.78 | 33.33 | 11.11 | 20.00 | 9.22 | 12.12 | 3.33 | 14.29 | 9.55 | 21.46 | 18.46 | 3.41 | 3.81 | 2.61 | 2.38 | 3.70 | 2.14 | 3.73 | 8.89 | 0.00 | 3.13 | 3.73 | 0.00 | 2.13 | 0.00 | 2.38 |
| Claude-3.5-Sonnet | 20.42 | 28.35 | 48.15 | 15.28 | 0.00 | 13.48 | 15.28 | 13.33 | 17.86 | 11.96 | 25.67 | 20.00 | 5.68 | 7.62 | 2.61 | 4.76 | 3.70 | 2.14 | 1.49 | 4.44 | 0.00 | 3.13 | 2.90 | 1.49 | 8.51 | 0.00 | 2.38 |
| Claude-3.7-Sonnet | 32.57 | 61.86 | 44.44 | 20.83 | 0.00 | 10.64 | 13.64 | 20.00 | 28.57 | 17.67 | 40.61 | 27.69 | 9.09 | 7.62 | 4.58 | 2.38 | 7.41 | 3.06 | 8.51 | 11.11 | 0.00 | 3.13 | 3.73 | 1.49 | 8.51 | 0.00 | 0.00 |
| DeepSeek-V3 | 8.80 | 7.22 | 33.33 | 5.56 | 10.00 | 9.22 | 10.61 | 0.00 | 10.71 | 4.17 | 2.68 | 9.23 | 2.27 | 3.81 | 3.27 | 2.38 | 3.70 | 1.38 | 0.75 | 6.67 | 0.00 | 2.07 | 0.75 | 0.75 | 2.27 | 0.00 | 0.00 |
| DeepSeek-R1 | 6.16 | 2.58 | 14.81 | 9.72 | 10.00 | 6.38 | 4.55 | 3.33 | 17.86 | 2.74 | 1.92 | 12.31 | 6.82 | 1.90 | 0.00 | 4.76 | 3.70 | 10.17 | 2.07 | 6.67 | 0.00 | 2.07 | 2.07 | 0.75 | 2.13 | 0.00 | 2.38 |
| Qwen2.5-72B-Instruct | 6.16 | 15.46 | 7.41 | 1.39 | 0.00 | 1.42 | 1.52 | 3.33 | 10.71 | 1.87 | 1.92 | 1.54 | 1.90 | 1.90 | 0.00 | 0.79 | 0.00 | 0.46 | 0.41 | 4.44 | 0.00 | 0.00 | 0.83 | 0.00 | 2.13 | 0.00 | 0.00 |
| Doubao-1.5-pro | 9.68 | 17.53 | 11.11 | 2.78 | 10.00 | 5.67 | 1.52 | 3.33 | 17.86 | 4.72 | 10.73 | 7.69 | 2.27 | 1.90 | 0.65 | 0.79 | 3.70 | 0.15 | 0.83 | 0.00 | 2.78 | 3.13 | 0.41 | 0.00 | 4.26 | 0.00 | 0.00 |
| Doubao-1.5-thinking | 20.60 | 41.24 | 22.22 | 11.11 | 0.00 | 9.93 | 7.58 | 3.33 | 10.71 | 10.54 | 26.82 | 13.85 | 6.82 | 1.90 | 2.61 | 1.59 | 5.08 | 1.53 | 1.66 | 6.67 | 0.00 | 0.00 | 1.66 | 2.13 | 6.38 | 0.00 | 0.00 |
| Gemini-2.5-Pro | 27.11 | 38.66 | 59.26 | 19.44 | 20.00 | 19.15 | 15.15 | 13.33 | 21.43 | 15.59 | 23.75 | 32.31 | 5.68 | 16.19 | 8.50 | 8.73 | 8.47 | 2.45 | 3.32 | 4.44 | 0.00 | 1.56 | 3.32 | 6.38 | 6.38 | 0.00 | 0.00 |
| Llama-4-Maverick | 6.87 | 3.09 | 29.63 | 8.33 | 20.00 | 5.67 | 4.55 | 0.00 | 21.43 | 3.18 | 1.53 | 18.46 | 4.55 | 0.95 | 1.31 | 2.38 | 0.00 | 1.07 | 2.07 | 0.00 | 0.00 | 0.00 | 2.07 | 0.00 | 0.00 | 0.00 | 0.00 |
| **MopenHands** | | | | | | | | | | | | | | | | | | | | | | | | | | | |
| GPT-4o | 17.96 | 38.66 | 29.63 | 0.00 | 0.00 | 8.51 | 6.06 | 0.00 | 10.71 | 1.07 | 4.44 | 6.15 | 0.00 | 2.86 | 1.96 | 2.38 | 0.00 | 3.39 | 0.00 | 4.44 | 0.00 | 0.00 | 1.66 | 0.00 | 2.13 | 0.00 | 0.00 |
| OpenAI-o1 | 10.56 | 18.56 | 7.41 | 1.39 | 30.00 | 8.51 | 3.03 | 6.67 | 7.14 | 0.92 | 2.22 | 4.62 | 0.00 | 6.67 | 1.96 | 2.38 | 3.70 | 5.08 | 1.24 | 2.22 | 0.00 | 0.00 | 1.24 | 0.75 | 0.00 | 0.00 | 0.00 |
| OpenAI-o3-mini-high | 16.55 | 31.44 | 22.22 | 1.39 | 40.00 | 4.96 | 13.64 | 6.67 | 14.29 | 1.53 | 4.44 | 10.77 | 3.64 | 3.81 | 1.96 | 0.79 | 6.78 | 1.69 | 0.00 | 6.67 | 0.00 | 0.00 | 1.66 | 0.00 | 0.00 | 0.00 | 2.38 |
| Claude-3.5-Sonnet | 28.35 | 48.97 | 25.93 | 18.06 | 10.00 | 12.77 | 24.24 | 6.67 | 32.14 | 2.60 | 6.67 | 18.46 | 0.00 | 3.81 | 5.23 | 7.14 | 10.17 | 6.78 | 2.24 | 13.33 | 0.00 | 1.56 | 2.24 | 0.00 | 0.00 | 0.00 | 2.38 |
| Claude-3.7-Sonnet | 35.21 | 71.65 | 48.15 | 2.78 | 30.00 | 11.35 | 21.21 | 13.33 | 32.14 | 3.68 | 9.11 | 23.08 | 2.27 | 7.62 | 9.15 | 13.49 | 15.25 | 10.17 | 2.90 | 11.11 | 0.00 | 2.90 | 2.90 | 1.49 | 14.89 | 2.27 | 2.38 |
| DeepSeek-V3 | 17.78 | 41.24 | 18.52 | 2.78 | 30.00 | 2.13 | 6.06 | 6.67 | 17.86 | 1.23 | 6.67 | 21.46 | 0.00 | 1.90 | 0.00 | 3.97 | 3.70 | 8.47 | 0.83 | 6.67 | 0.00 | 0.00 | 0.83 | 0.00 | 4.26 | 0.00 | 0.00 |
| DeepSeek-R1 | 17.78 | 41.24 | 14.81 | 1.39 | 10.00 | 2.13 | 13.64 | 6.67 | 14.29 | 0.77 | 6.67 | 19.16 | 0.00 | 2.86 | 0.00 | 1.59 | 5.08 | 3.39 | 2.07 | 6.67 | 0.00 | 2.07 | 2.07 | 0.00 | 0.00 | 0.00 | 0.00 |
| Qwen2.5-72B-Instruct | 4.23 | 6.70 | 7.41 | 0.00 | 20.00 | 13.64 | 1.52 | 0.00 | 10.71 | 0.46 | 0.00 | 10.77 | 0.00 | 0.00 | 1.31 | 1.59 | 0.00 | 3.39 | 0.00 | 0.00 | 0.00 | 0.00 | 0.00 | 0.00 | 4.26 | 0.00 | 0.00 |
| Doubao-1.5-pro | 7.57 | 15.46 | 0.00 | 0.00 | 10.00 | 4.96 | 1.52 | 0.00 | 14.29 | 0.61 | 2.22 | 3.45 | 0.00 | 0.95 | 0.00 | 0.79 | 0.00 | 0.00 | 0.75 | 2.22 | 0.00 | 0.83 | 0.00 | 0.75 | 4.26 | 0.00 | 0.00 |
| Doubao-1.5-thinking | 24.30 | 42.78 | 33.33 | 11.11 | 20.00 | 2.13 | 7.58 | 13.33 | 21.43 | 2.45 | 6.67 | 7.69 | 4.55 | 8.10 | 2.61 | 7.94 | 1.85 | 1.69 | 2.99 | 2.22 | 0.00 | 0.00 | 5.39 | 2.99 | 8.51 | 0.00 | 2.38 |
| Gemini-2.5-Pro | 39.08 | 63.92 | 22.22 | 33.33 | 40.00 | 25.53 | 31.82 | 10.00 | 22.28 | 5.51 | 9.20 | 15.38 | 22.73 | 33.33 | 9.15 | 9.15 | 11.86 | 1.86 | 8.51 | 4.44 | 0.00 | 9.38 | 7.88 | 2.99 | 8.51 | 0.00 | 2.38 |
| Llama-4-Maverick | 16.20 | 23.71 | 18.52 | 15.28 | 30.00 | 12.06 | 7.58 | 10.00 | 7.14 | 2.14 | 4.44 | 4.62 | 6.82 | 6.67 | 3.92 | 3.17 | 1.85 | 3.39 | 0.75 | 4.44 | 0.00 | 1.56 | 2.90 | 0.75 | 6.38 | 4.55 | 0.00 |

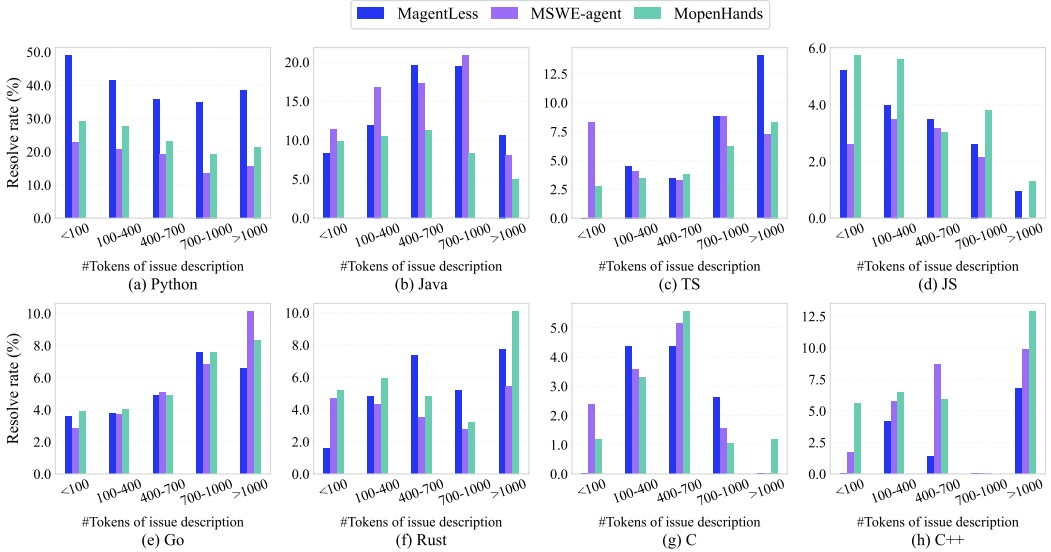

Figure 11: Influence of issue description length on resolved rate.

potential types of long issue descriptions: (1) detailed issues with precise issue position indications and resolving steps, and (2) complex issues that require extended descriptions to explain. These two possibilities have distinct impacts on the difficulty of resolving an issue, influencing the resolved rate in different ways.

## E.6 Influence of Fix Patch

In this subsection, we investigate the impact of the ground-truth fix patches on the resolved rate, focusing on two key factors: (1) *Fix patch length*: We analyze how the length of fix patches affects performance, noticing that longer patches require more complex reasoning capabilities from LLMs. The fix patches are categorized into five intervals based on the length distribution shown in Fig. **??**: <200, 200-600, 600-1000, 1000-1400, and >1400 tokens. (2) *Number of files modified by fix patches*: We examine how the cross-file nature of the fix patches influences performance, with more files requiring enhanced cross-file handling capabilities. The number of modified files is divided into four categories: 1, 1-5, 5-10, and >10, with the distribution shown in Fig.**??**. The detailed results across various programming languages are shown in Fig. 12 and Fig. 13, respectively. The absence of corresponding bars indicates cases where no issues are successfully resolved.

## E.7 Case Study

In this subsection, we analyze representative cases that highlight the strengths of agents, common failure patterns, and language-specific challenges, providing insights for future directions.

### E.7.1 Language-General Case

- MSWE-agent and MopenHands often failed by exhausting the 50-round interaction limit, sometimes without even triggering the submit action, as seen in cases like axios__axios-5919.traj, clap-rs__clap-5520.traj, and cli__cli-513.traj. Future work may explore strategies that enable agents to solve more complex tasks within a limited number of interaction rounds.

- A significant number of failures across all three agent methods were due to incorrect fault localization, which led to an inability to identify and modify the relevant code, as seen in cases such as elastic__logstash-14898.traj, alibaba__fastjson2-2285.traj, fasterxml__jackson-databind-3560.traj, and apache__dubbo-7041.traj. This highlights the centrality of accurate fault localization and points to the potential of integrating software engineering techniques like SBFL [2, 21] into future agent designs.

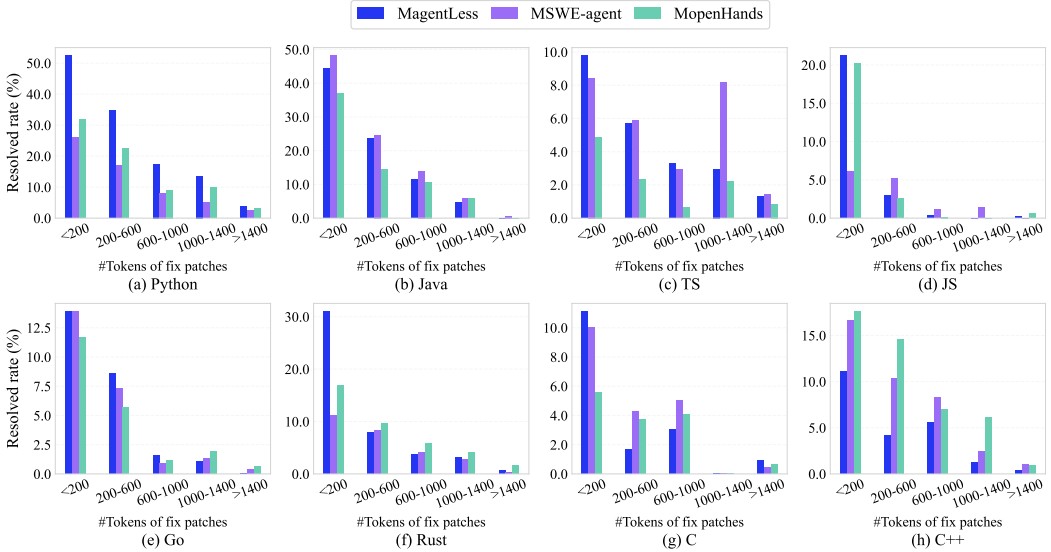

Figure 12: Influence of fix patch length on resolved rate.

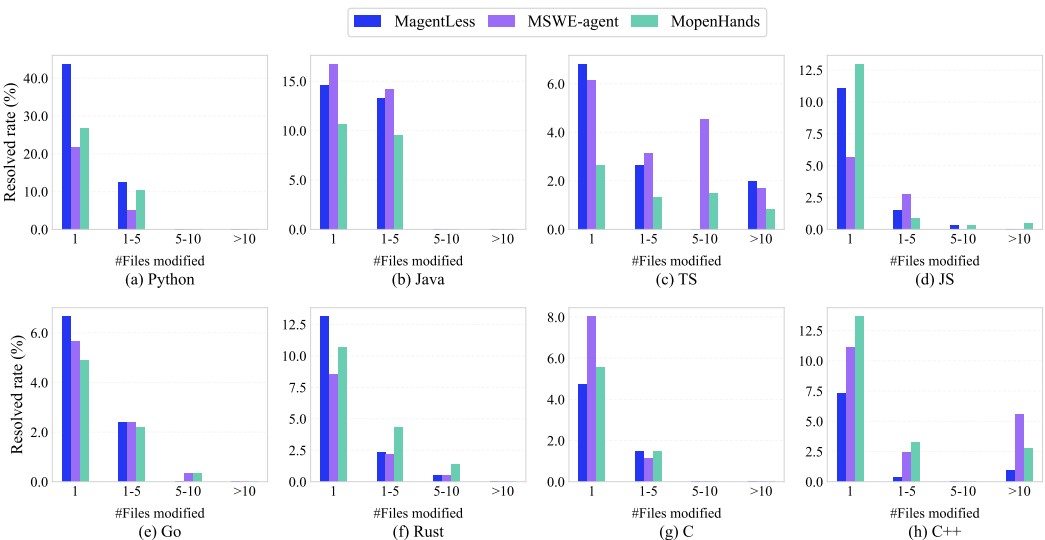

Figure 13: Influence of the number of files modified by fix patches on resolved rate.

- In cases such as astropy__astropy-12907.traj and django__django-11299.traj, the model generated multiple valid actions in a single turn, but the hardcoded agent framework executed only the last, resulting in premature submission. This reveals a structural bottleneck in current agent design, where rigid control logic overrides model intent. It calls for a shift toward lightweight, model-centric agents with full decision autonomy delegated to the LLM.

- Bug reproduction plays a critical role in successful repair. In cases such as nlohmann__json-4537.traj, fmtlib__fmt-3248.traj, fasterxml__jackson-core-1142.traj, and google__gson-1093.traj, the model successfully reproduced the issue before producing an effective fix. In contrast, failure to reproduce often resulted in unresolved cases, as seen in catchorg__Catch2-1609.traj. However, reproduction is not always a prerequisite for success. Claude-3.5-Sonnet and Claude-3.7-Sonnet occasionally bypass reproduction and edit the code directly—yet still resolve the issue successfully, as in nlohmann__json-3601.traj, fmtlib__fmt-3729.traj, and googlecontainertools__jib-4035.traj. These cases suggest that agents should selectively invoke reproduction based on factors such as error traceability, edit confidence, and execution cost.

### E.7.2 Language-Specific Case

- For certain TypeScript projects, the length of the extracted repository structure often exceeds the model's maximum context length, preventing MagentLess from performing fault localization (e.g., mui__material-ui-25852.traj and mui__material-ui-37850.traj). This reveals the limited generalizability of fixed workflows like MagentLess when confronted with structurally irregular and language-specific scenarios, indicating significant room for improvement in both robustness and adaptability.

- Tree-sitter fails to reliably extract code structures in JavaScript repositories that use loosely bound syntax such as arrow functions, preventing MagentLess from constructing contextual windows around candidate edits (e.g., iamkun__dayjs-2532.traj and iamkun__dayjs-2399.traj). This exposes a structural brittleness in syntax-driven workflows when applied to syntactically permissive languages, motivating future extensions of MagentLess toward greater tolerance to parsing failure and language-specific irregularities.

- In some JavaScript projects, agents sometimes invoke `pnpm` to launch development servers as part of the repair routine. However, current agent frameworks lack support for managing long-lived, interactive processes, often resulting in premature termination or container crashes (e.g., sveltejs__svelte-12460.traj and sveltejs__svelte-10077.traj). Future agents should support persistent shell sessions and interactive service control, as enabled by frameworks like SWE-ReX [34].

### E.8 Token Consumption

Table 12: Average cost ($) per issue of different models and methods on Multi-SWE-bench.

| Methods | Models | Python | Java | TS | JS | Go | Rust | C | C++ |
|---|---|---|---|---|---|---|---|---|---|
| MagentLess | GPT-4o | 0.1324 | 0.1576 | 0.6230 | 0.0990 | 0.0900 | 0.1476 | 0.1520 | 0.2153 |
| | OpenAI-o1 | 0.7417 | 0.8680 | 3.6795 | 0.6233 | 0.4698 | 0.9682 | 0.8153 | 1.8734 |
| | OpenAI-o3-mini-high | 0.0543 | 0.0978 | 0.2767 | 0.0489 | 0.0421 | 0.0847 | 0.0896 | 0.2287 |
| | Claude-3.5-Sonnet | 0.1981 | 0.1853 | 0.7478 | 0.1213 | 0.1102 | 0.1937 | 0.1856 | 0.3280 |
| | Claude-3.7-Sonnet | 0.1821 | 0.2393 | 0.7763 | 0.1275 | 0.1229 | 0.2933 | 0.1994 | 0.4317 |
| | DeepSeek-V3 | 0.0075 | 0.0059 | 0.0192 | 0.0046 | 0.0045 | 0.0085 | 0.0091 | 0.0156 |
| | DeepSeek-R1 | 0.0105 | 0.0137 | 0.0373 | 0.0068 | 0.0070 | 0.0158 | 0.0172 | 0.0280 |
| | Qwen2.5-72B-Instruct | 0.0051 | 0.0092 | 0.0324 | 0.0042 | 0.0046 | 0.0077 | 0.0084 | 0.0204 |
| | Doubao-1.5-pro | 0.0055 | 0.0132 | 0.0279 | 0.0046 | 0.0041 | 0.0142 | 0.0138 | 0.0240 |
| | Doubao-1.5-thinking | 0.0251 | 0.0381 | 0.1412 | 0.0198 | 0.0188 | 0.0399 | 0.0438 | 0.1185 |
| | Gemini-2.5-Pro | 0.1511 | 0.1130 | 0.3371 | 0.0735 | 0.1012 | 0.1092 | 0.1018 | 0.2431 |
| | Llama-4-Maverick | 0.0140 | 0.0152 | 0.0534 | 0.0109 | 0.0110 | 0.0196 | 0.0158 | 0.0311 |
| MSWE-agent | GPT-4o | 0.4480 | 0.1731 | 0.1623 | 0.1236 | 0.1390 | 0.1565 | 0.1444 | 0.1883 |
| | OpenAI-o1 | 3.7499 | 0.6797 | 0.6644 | 0.5772 | 0.7749 | 0.8151 | 0.7010 | 0.6353 |
| | OpenAI-o3-mini-high | 0.2722 | 0.0450 | 0.0350 | 0.0410 | 0.0441 | 0.0538 | 0.0422 | 0.0572 |
| | Claude-3.5-Sonnet | 0.1831 | 0.1546 | 0.1110 | 0.1266 | 0.1091 | 0.1669 | 0.1451 | 0.1794 |
| | Claude-3.7-Sonnet | 0.1626 | 0.1887 | 0.1700 | 0.1654 | 0.1698 | 0.1901 | 0.1803 | 0.1810 |
| | DeepSeek-V3 | 0.0260 | 0.0070 | 0.0035 | 0.0049 | 0.0037 | 0.0084 | 0.0034 | 0.0068 |
| | DeepSeek-R1 | 0.0075 | 0.0083 | 0.0050 | 0.0066 | 0.0055 | 0.0082 | 0.0069 | 0.0088 |
| | Qwen2.5-72B-Instruct | 0.0241 | 0.0106 | 0.0083 | 0.0072 | 0.0063 | 0.0079 | 0.0061 | 0.0134 |
| | Doubao-1.5-pro | 0.0083 | 0.0052 | 0.0028 | 0.0046 | 0.0039 | 0.0053 | 0.0042 | 0.0046 |
| | Doubao-1.5-thinking | 0.0329 | 0.0334 | 0.0194 | 0.0217 | 0.0187 | 0.0353 | 0.0284 | 0.0400 |
| | Gemini-2.5-Pro | 0.1215 | 0.0862 | 0.0783 | 0.0971 | 0.0794 | 0.1208 | 0.1044 | 0.1046 |
| | Llama-4-Maverick | 0.0088 | 0.0094 | 0.0077 | 0.0076 | 0.0066 | 0.0092 | 0.0066 | 0.0090 |
| MopenHands | GPT-4o | 0.0758 | 0.0682 | 0.1054 | 0.1038 | 0.0751 | 0.1155 | 0.1078 | 0.1031 |
| | OpenAI-o1 | 0.3608 | 0.3564 | 0.5374 | 0.5885 | 0.4099 | 0.5262 | 0.5081 | 0.4171 |
| | OpenAI-o3-mini-high | 0.0465 | 0.0422 | 0.0528 | 0.0581 | 0.0462 | 0.0476 | 0.0407 | 0.0449 |
| | Claude-3.5-Sonnet | 0.2124 | 0.1761 | 0.2041 | 0.2089 | 0.1908 | 0.2601 | 0.2523 | 0.2086 |
| | Claude-3.7-Sonnet | 0.1957 | 0.2032 | 0.2028 | 0.2261 | 0.2080 | 0.2500 | 0.2002 | 0.2158 |
| | DeepSeek-V3 | 0.0070 | 0.0059 | 0.0059 | 0.0069 | 0.0047 | 0.0079 | 0.0054 | 0.0080 |
| | DeepSeek-R1 | 0.0128 | 0.0134 | 0.0113 | 0.0141 | 0.0130 | 0.0177 | 0.0168 | 0.0191 |
| | Qwen2.5-72B-Instruct | 0.0077 | 0.0090 | 0.0084 | 0.0074 | 0.0073 | 0.0092 | 0.0084 | 0.0089 |
| | Doubao-1.5-pro | 0.0037 | 0.0036 | 0.0025 | 0.0034 | 0.0031 | 0.0036 | 0.0037 | 0.0038 |
| | Doubao-1.5-thinking | 0.0245 | 0.0198 | 0.0254 | 0.0259 | 0.0213 | 0.0268 | 0.0252 | 0.0286 |
| | Gemini-2.5-Pro | 0.1896 | 0.0848 | 0.1839 | 0.1891 | 0.1497 | 0.1705 | 0.2100 | 0.1732 |
| | Llama-4-Maverick | 0.0078 | 0.0080 | 0.0076 | 0.0079 | 0.0079 | 0.0100 | 0.0095 | 0.0094 |

Tab. 13 compares the average token consumption for various languages using the GPT-4o tokenizer. Overall, token consumption varies between methods and languages. Among languages, TS exhibits the highest token consumption in MagentLess, whereas Python is the most token-intensive language in MSWE-agent. Notably, Go demonstrates relatively low token consumption in both input and

Table 13: Average token consumption on Multi-SWE-bench. In. represents the average number of input tokens (in thousands), and Out. is the average number of output tokens (in thousands).

| Models | Python | | Java | | TS | | JS | | Go | | Rust | | C | | C++ | |
|---|---|---|---|---|---|---|---|---|---|---|---|---|---|---|---|---|
| | In. | Out. | In. | Out. | In. | Out. | In. | Out. | In. | Out. | In. | Out. | In. | Out. | In. | Out. |
| **MagentLess** | | | | | | | | | | | | | | | | |
| GPT-4o | 36.15 | 4.20 | 52.10 | 2.74 | 241.18 | 2.01 | 29.48 | 2.53 | 25.14 | 2.72 | 48.23 | 2.71 | 50.26 | 2.64 | 76.38 | 2.44 |
| OpenAI-o1 | 34.43 | 3.76 | 50.18 | 1.92 | 240.47 | 1.21 | 36.53 | 1.26 | 24.51 | 1.70 | 58.59 | 1.49 | 48.08 | 1.57 | 119.64 | 1.31 |
| OpenAI-o3-mini-high | 31.38 | 4.50 | 79.48 | 2.36 | 245.39 | 1.54 | 38.28 | 1.55 | 31.58 | 1.67 | 68.80 | 2.05 | 73.48 | 1.99 | 200.29 | 1.91 |
| Claude-3.5-Sonnet | 39.13 | 5.38 | 48.42 | 2.67 | 239.93 | 1.86 | 28.46 | 2.39 | 22.80 | 2.79 | 51.25 | 2.66 | 49.13 | 2.55 | 96.85 | 2.49 |
| Claude-3.7-Sonnet | 27.99 | 6.54 | 63.97 | 3.16 | 248.36 | 2.08 | 26.66 | 3.17 | 22.79 | 3.63 | 81.15 | 3.33 | 50.52 | 3.19 | 129.34 | 2.91 |
| DeepSeek-V3 | 39.97 | 4.26 | 42.35 | 2.70 | 244.32 | 1.92 | 26.44 | 2.51 | 22.78 | 2.65 | 83.04 | 2.47 | 92.53 | 2.38 | 189.08 | 2.11 |
| DeepSeek-R1 | 31.35 | 2.80 | 70.35 | 1.76 | 249.02 | 1.10 | 28.23 | 1.30 | 21.69 | 1.79 | 88.73 | 1.52 | 100.99 | 1.39 | 177.66 | 1.41 |
| Qwen2.5-72B-Instruct | 28.60 | 3.46 | 62.95 | 2.52 | 243.98 | 1.65 | 26.11 | 2.14 | 24.67 | 3.36 | 50.63 | 2.89 | 55.93 | 2.78 | 150.44 | 2.19 |
| Doubao-1.5-pro | 42.75 | 2.91 | 116.09 | 1.36 | 249.51 | 1.55 | 36.37 | 2.07 | 29.38 | 3.15 | 124.67 | 1.62 | 121.94 | 1.52 | 216.21 | 0.76 |
| Doubao-1.5-thinking | 31.76 | 3.31 | 61.34 | 1.72 | 248.36 | 0.96 | 29.62 | 1.47 | 24.70 | 2.24 | 64.31 | 1.76 | 71.87 | 1.61 | 206.95 | 1.17 |
| Gemini-2.5-Pro | 39.05 | 10.22 | 51.63 | 4.84 | 242.51 | 3.40 | 30.18 | 3.58 | 28.32 | 6.58 | 55.77 | 3.95 | 49.51 | 3.99 | 169.89 | 3.07 |
| Llama-4-Maverick | 35.54 | 8.07 | 48.49 | 6.46 | 243.05 | 5.61 | 30.95 | 5.53 | 26.25 | 6.79 | 71.47 | 6.28 | 54.09 | 5.90 | 128.45 | 6.35 |
| **MSWE-agent** | | | | | | | | | | | | | | | | |
| GPT-4o | 166.91 | 3.08 | 51.05 | 4.54 | 46.39 | 4.63 | 32.01 | 4.36 | 36.73 | 4.71 | 43.79 | 4.71 | 39.47 | 4.57 | 55.49 | 4.96 |
| OpenAI-o1 | 243.44 | 1.64 | 33.36 | 2.99 | 30.05 | 3.56 | 25.70 | 3.19 | 37.71 | 3.49 | 39.51 | 3.71 | 34.05 | 3.17 | 29.24 | 3.28 |
| OpenAI-o3-mini-high | 240.23 | 1.82 | 26.37 | 3.64 | 18.27 | 3.39 | 21.33 | 3.99 | 26.46 | 3.41 | 32.84 | 4.03 | 23.24 | 3.78 | 32.39 | 4.90 |
| Claude-3.5-Sonnet | 33.30 | 5.55 | 32.09 | 3.89 | 21.51 | 3.10 | 23.94 | 3.66 | 21.06 | 3.06 | 35.47 | 4.03 | 31.16 | 3.44 | 38.22 | 4.32 |
| Claude-3.7-Sonnet | 31.86 | 4.46 | 38.96 | 4.79 | 32.08 | 4.92 | 32.16 | 4.60 | 33.79 | 4.56 | 40.59 | 4.56 | 38.41 | 4.34 | 36.96 | 4.67 |
| DeepSeek-V3 | 12.63 | 22.83 | 35.08 | 4.14 | 15.73 | 2.15 | 19.78 | 3.23 | 15.34 | 2.43 | 33.98 | 5.47 | 16.26 | 2.07 | 31.28 | 4.18 |
| DeepSeek-R1 | 11.76 | 2.65 | 17.51 | 2.69 | 9.91 | 1.66 | 9.36 | 2.43 | 10.47 | 1.85 | 13.98 | 2.86 | 11.34 | 2.44 | 14.64 | 3.06 |
| Qwen2.5-72B-Instruct | 164.42 | 6.69 | 53.43 | 9.26 | 39.58 | 7.82 | 35.21 | 6.45 | 22.53 | 8.38 | 36.49 | 7.93 | 28.90 | 5.76 | 67.29 | 11.69 |
| Doubao-1.5-pro | 72.58 | 1.30 | 37.75 | 3.73 | 19.18 | 2.46 | 32.90 | 3.68 | 25.39 | 3.91 | 38.09 | 4.04 | 29.03 | 3.65 | 32.67 | 3.59 |
| Doubao-1.5-thinking | 47.57 | 2.86 | 35.32 | 6.19 | 21.06 | 3.47 | 21.91 | 4.27 | 19.06 | 3.65 | 37.47 | 6.53 | 30.49 | 5.14 | 43.63 | 7.09 |
| Gemini-2.5-Pro | 25.81 | 8.92 | 30.79 | 4.78 | 25.76 | 4.61 | 32.57 | 5.64 | 25.22 | 4.79 | 45.26 | 6.43 | 39.10 | 5.55 | 36.99 | 5.84 |
| Llama-4-Maverick | 21.32 | 5.31 | 32.41 | 3.41 | 25.68 | 3.07 | 24.55 | 3.14 | 20.66 | 2.90 | 32.13 | 3.31 | 21.96 | 2.62 | 31.90 | 3.11 |
| **MopenHands** | | | | | | | | | | | | | | | | |
| GPT-4o | 25.35 | 1.24 | 22.01 | 1.32 | 35.76 | 1.60 | 35.51 | 1.50 | 23.96 | 1.52 | 40.40 | 1.45 | 34.80 | 2.08 | 34.61 | 1.66 |
| OpenAI-o1 | 19.27 | 1.20 | 18.69 | 1.27 | 27.28 | 2.14 | 30.96 | 2.07 | 21.09 | 1.56 | 28.90 | 1.55 | 27.18 | 1.67 | 21.55 | 1.57 |
| OpenAI-o3-mini-high | 21.52 | 5.18 | 22.82 | 3.88 | 30.70 | 4.32 | 36.57 | 4.06 | 25.44 | 4.14 | 30.64 | 3.15 | 23.98 | 3.26 | 23.76 | 4.26 |
| Claude-3.5-Sonnet | 32.35 | 7.69 | 31.97 | 5.35 | 35.88 | 6.43 | 38.91 | 6.14 | 27.31 | 7.26 | 55.51 | 6.23 | 55.79 | 5.66 | 35.85 | 6.74 |
| Claude-3.7-Sonnet | 26.04 | 7.84 | 28.43 | 7.86 | 31.06 | 7.31 | 38.06 | 7.46 | 30.05 | 7.86 | 48.30 | 7.00 | 35.25 | 6.30 | 33.14 | 7.76 |
| DeepSeek-V3 | 18.97 | 5.16 | 26.35 | 3.65 | 26.60 | 3.69 | 29.08 | 4.43 | 15.42 | 3.31 | 32.90 | 5.05 | 21.77 | 3.53 | 30.67 | 5.36 |
| DeepSeek-R1 | 11.25 | 5.13 | 17.15 | 5.04 | 12.71 | 4.33 | 17.85 | 5.29 | 12.65 | 5.14 | 17.58 | 6.95 | 24.16 | 6.11 | 17.38 | 7.62 |
| Qwen2.5-72B-Instruct | 27.28 | 10.38 | 33.26 | 11.80 | 36.86 | 9.12 | 28.84 | 9.07 | 21.17 | 11.35 | 37.14 | 10.99 | 35.02 | 9.69 | 35.34 | 10.88 |
| Doubao-1.5-pro | 23.16 | 3.95 | 24.15 | 3.35 | 18.34 | 1.66 | 23.75 | 2.76 | 18.21 | 3.78 | 27.40 | 2.07 | 26.54 | 2.82 | 26.07 | 3.44 |
| Doubao-1.5-thinking | 17.71 | 6.63 | 18.58 | 4.25 | 26.50 | 4.79 | 23.79 | 5.72 | 15.89 | 5.64 | 24.20 | 6.01 | 24.53 | 5.21 | 23.62 | 7.00 |
| Gemini-2.5-Pro | 27.38 | 15.54 | 20.39 | 5.93 | 36.15 | 13.87 | 36.26 | 14.38 | 30.59 | 11.15 | 43.45 | 11.61 | 45.97 | 15.25 | 35.81 | 12.84 |
| Llama-4-Maverick | 22.12 | 3.96 | 21.93 | 4.24 | 21.15 | 3.91 | 22.27 | 4.00 | 21.85 | 4.15 | 31.64 | 4.36 | 31.34 | 3.86 | 29.01 | 4.27 |

output, likely due to its minimalistic syntax and clear conventions, which contribute to its compact representation and reduced token overhead. Additionally, in MSWE-agent for Python, we observe increased token usage on LLMs, including GPT-4o, OpenAI-o1, OpenAI-o3-mini-high, and Qwen2.5-72B-Instruct. This is because we maintain the original SWE-agent implementation for Python, which does not incorporate the over-length truncation mechanism applied to other languages.

# F   Potential Societal Impacts

Multi-SWE-bench can positively impact software engineering by improving automated issue resolution across multiple programming languages, leading to faster and more reliable software development. However, it may also lead to job displacement in certain areas of software maintenance and introduce risks if models generate errors or are misused for malicious purposes. To mitigate these risks, we advocate for responsible use, transparency, and continuous monitoring.

