# OpenReview forum: "Multi-SWE-bench: A Multilingual Benchmark for Issue Resolving"
_NeurIPS.cc/2025/Datasets_and_Benchmarks_Track — NeurIPS 2025 Datasets and Benchmarks Track poster_

### Official Review · Reviewer_3v4u · 2025-06-29

**Rating:** 5
**Confidence:** 4

**Summary:**

This paper introduces Multi-SWE-bench, a multilingual benchmark for evaluating issue resolution in code, covering 8 major programming languages with 2,132 high-quality, human-verified GitHub issues. The dataset is constructed via a rigorous five-phase pipeline  and includes a time-based difficulty stratification. Experiments evaluate 12 state-of-the-art LLMs  across three methods (Agentless, SWE-agent, OpenHands), revealing three key findings:

1. Limited cross-language generalization: Strong performance on Python but significant drops in other languages.
2. Performance aligns with human-labeled difficulty: Resolution rates decrease sharply for harder issues.
3. Severe degradation on cross-file patches: Near-zero success when modifications span multiple files.

The authors open-source the dataset, code, and Docker images, while also proposing Multi-SWE-bench Mini (a lightweight subset) and Multi-SWE-RL  (a reinforcement learning dataset).

**Dataset Code Accessibility:**

Yes

**Dataset Code Comments:**

The paper provides sufficient access and details about the datasets.

**Ethical Considerations:**

No, there are no or only very minor ethics concerns

**Final Justification:**

The author rebuttal addresses my concerns in the initial reviews. I believe this work is a good complementary to SWE-Bench from the multilingual perspective.

**Limitations Weaknesses:**

1. Over-reliance on "Resolved Rate": No complementary metrics (e.g., patch correctness, localization accuracy) to assess quality beyond binary success.
2. Uneven language distribution: JavaScript (246) and Rust (239) have more samples than C++ (129) and C (148), potentially skewing cross-language comparisons.
3. Superficial analysis of cross-language gaps: Attributes performance drops to "language complexity" (e.g., async, memory management) without probing deeper causes (e.g., pretraining data bias, toolchain support).

**Strengths Contributions:**

1. The first large-scale multilingual benchmark for issue resolution, overcoming the Python-centric bias of prior work (e.g., SWE-bench). Enables evaluation of LLMs in real-world, cross-language, and multi-file scenarios, pushing toward more robust AI programmers.
2. Five-phase pipeline ensures reliability: High-star repositories, reproducible Docker environments, regression-aware filtering, and dual expert annotation  (68 annotators). Difficulty stratification based on actual fix time (human-aligned) provides fine-grained insights into model capabilities.
3. Tests 12 LLMs across three methods, analyzing performance by language, difficulty, and issue type (bug fixes, new features, optimizations). Reveals language-specific challenges (e.g., async runtime in TS/JS, manual memory management in C/C++), guiding future research.
4. Quantifies the impact of issue description length, patch size, and cross-file edits on resolution rates (e.g., >600-token patches lead to sharp drops).
5. Fully releases dataset, code, and Docker images, ensuring reproducibility. Provides Multi-SWE-bench Mini (400 samples) and Multi-SWE-RL (4,723 instances) to lower entry barriers.

---

> ### Author Rebuttal · Authors · 2025-07-28
>
> We are grateful for your endorsement of Multi-SWE-bench. Regarding the limitations you raised, we will provide detailed responses to address your concerns.
>
> __Q1: Over-reliance on "Resolved Rate": No complementary metrics (e.g., patch correctness, localization accuracy) to assess quality beyond binary success.__
>
> A1: Thank you for raising this concern. In fact, we have included localization accuracy (Section 5 and Appendix Figure 5), cost (Appendix Table 12), and token consumption (Appendix Table 13) as complementary metrics to assess quality beyond binary success.
> We did not use patch correctness you mentioned (which we understand as the success rate of applying patches) because current powerful LLMs can achieve nearly 100% success rates, making it non-discriminative.
>
> Overall, we think that test case-based resolved rate is the most direct and effective metric, while we also incorporate all meaningful metrics we could identify as complementary metrics.
>
> __Q2: Uneven language distribution: JavaScript (246) and Rust (239) have more samples than C++ (129) and C (148), potentially skewing cross-language comparisons.__
>
> A2: Thank you for raising this concern. We have addressed this through the _Mini_ version of Multi-SWE-bench (Appendix Section E.1), which has an even language distribution: 50 instances for each of the 8 languages, totaling 400 instances.
>
> __Q3: Superficial analysis of cross-language gaps: Attributes performance drops to "language complexity" (e.g., async, memory management) without probing deeper causes (e.g., pretraining data bias, toolchain support).__
>
> A3: We totally agree with this. Indeed, one of the original goals of Multi-SWE-bench was to investigate the root causes of cross-language performance gaps.
>
> Unfortunately, our analysis revealed that this goal involves numerous influencing factors (e.g., pretraining data bias, toolchain, agent frameworks, and model parameters), and all of which remain in a phase of rapid evolution without reaching convergence. Under such circumstances, these factors are not only interrelated but also exhibit synergistic effects, making it extremely challenging to draw solid conclusions at this stage.
> Therefore, we conducted our analysis of cross-language gaps to a preliminary level.
> But we expect to progressively uncover the root causes of this issue in the future through the ongoing maintenance and development of Multi-SWE-bench.
>
> We hope our answers have resolved some of your concerns. If so, we would appreciate it if you could increase the rating to 6. Thanks again for your constructive feedback and suggestions.

---

> > ### Comment · Reviewer_3v4u · 2025-08-02
> > **Thanks for the rebuttal**
> >
> > The rebuttal addresses most of my concerns. However, I believe 5 is a reasonable score for this paper. Good luck!

---

### Official Review · Reviewer_11BR · 2025-06-30

**Rating:** 5
**Confidence:** 5

**Summary:**

SWE-bench is the leading repo-level code benchmark, but it's just in Python. The SWE-bench team put out a Multilingual version, 9 langs, but it has just 300 issues.

This benchmark is 2k+ issues in 8 languages, all of which have been manually Verified as in SWE-bench Verified. They have lots of results from the 3 top baselines for these tasks : SWE-agent, OpenHands, and Agentless.

This paper is very thorough, the conclusions are interesting and the artifact will be useful for years to come. I support accepting this work.

**Additional Feedback:**

Can you talk about the differences between the SWE-bench scaffold and your scaffold- if an organization is already used to evaluating on SWE-bench, how many changes do they have to make to start evaluating on your benchmark?

**Dataset Code Accessibility:**

Yes

**Ethical Considerations:**

No, there are no or only very minor ethics concerns

**Final Justification:**

This is an interesting paper and artifact and I would like to see it getting accepted.

**Strengths Contributions:**

1. First large scale (2k+) SWE-bench for 8 languages.
2. Issues have been verified by human annotators as being solvable.
3. Very thorough agent baselines
4. This benchmark will be very useful in evaluating the real-world abilities of software engineering agents for years to come.
5. The low performance on the non-python langs shows the need for this benchmark (and is also super interesting to see).

---

> ### Author Rebuttal · Authors · 2025-07-28
>
> We greatly appreciate your highly positive assessment of Multi-SWE-bench. Below we answer your specific question:
>
> __Q: Can you talk about the differences between the SWE-bench scaffold and your scaffold- if an organization is already used to evaluating on SWE-bench, how many changes do they have to make to start evaluating on your benchmark?__
>
> A: There are no differences between SWE-bench's and Multi-SWE-bench's scaffolds. Therefore, to use it, no changes are needed: clone [our repository](https://github.com/multi-swe-bench/multi-swe-bench) and run `make install` to get started.
>
> Thanks again for your high recognition of our work.

---

### Official Review · Reviewer_C7SD · 2025-07-02

**Rating:** 5
**Confidence:** 4

**Summary:**

This paper introduces Multi-SWE-bench, a large-scale, multilingual benchmark for evaluating the ability of Large Language Models (LLMs) to resolve real-world software issues. The primary contribution is a high-quality dataset of 2,132 human-validated instances across eight major programming languages (Python, Java, C++, etc.), addressing the limitation of existing benchmarks that focus almost exclusively on Python. A key innovation is the human-aligned difficulty stratification (Easy, Medium, Hard), which enables a more nuanced evaluation of model capabilities. Through extensive experiments with 12 LLMs and 3 methods, the paper reveals critical limitations of current models, including poor generalization beyond Python, a sharp performance decline on complex tasks, and significant challenges with cross-file issue resolution. The entire benchmark, along with code and reproducible Docker environments, is open-sourced to facilitate future research.

**Dataset Code Accessibility:**

Yes

**Ethical Considerations:**

No, there are no or only very minor ethics concerns

**Final Justification:**

I maintain my rate

**Limitations Weaknesses:**

1. The automated PR filtering in Phase 4 requires at least one test case to transition from ANY -> FAILED -> PASSED. This definition inherently biases the dataset towards bug-fixing scenarios where a failing test already exists. This might limit the benchmark's suitability for evaluating performance on "new feature" or "feature optimization" tasks, where such pre-existing failing tests are less common. The authors acknowledge these categories but could provide a more detailed discussion on how the filtering methodology might affect the distribution and nature of these tasks in the final dataset.

2. The paper adapts existing agent frameworks for multilingual support. However, the core agent-computer interface (ACI) and tools (e.g., file system operations, text search) remain largely language-agnostic. The paper could benefit from a deeper discussion on whether this generic toolset is sufficient for languages with complex build systems, type systems, or debugging requirements (e.g., C++, Rust). The case studies touch upon this (e.g., Tree-sitter failures in JS ), but a more systematic analysis of tool-language fit would be a valuable addition.

3. The experiments show that the performance on "Hard" issues is near-zero for almost all models and methods. While this is a crucial finding, it also means this difficulty level currently acts as a ceiling, making it hard to differentiate between the most capable models. The benchmark could be improved in the future by further subdividing the "Hard" category (e.g., based on the 1-4 hour vs. >4 hour estimates from the raw annotations ) to create a more challenging gradient for next-generation models.

**Strengths Contributions:**

1. The paper addresses a clear and significant gap in existing research. While benchmarks like SWE-bench have driven progress, their Python-only focus provides a limited view of LLM capabilities. Multi-SWE-bench is the first large-scale, human-verified benchmark to offer broad multilingual coverage, making it an invaluable resource for assessing the true generalization ability of code agents.

2. The five-phase data construction pipeline is exceptionally rigorous and well-documented. The use of individualized, reproducible Docker environments for each instance is a critical strength that ensures high fidelity and reliability in evaluation.

3. he introduction of a time-based, human-aligned difficulty stratification (Easy, Medium, Hard) is a novel and highly valuable contribution. This allows for a much more fine-grained analysis of model performance than a single aggregate score.

4. The authors conducted a large-scale evaluation of 12 state-of-the-art LLMs using 3 representative methods, providing a comprehensive snapshot of the current landscape.

---

> ### Author Rebuttal · Authors · 2025-07-28
>
> Thanks for your highly positive reviews and thought-provoking feedback. We will incorporate your valuable suggestions, including (1) analyzing task-type distribution biases, (2) improving tool-language compatibility, and (3) establishing finer-grained difficulty categorization, to guide the ongoing maintenance and evolution of Multi-SWE-bench.

---

### Official Review · Reviewer_MxN5 · 2025-07-02

**Rating:** 5
**Confidence:** 4

**Summary:**

The current work introduces Multi-SWE-bench, a code-based benchmark to test the performance of LLM-based agentic modes on code issue resolving.

It builds over previous similar benchmarks such as SWE-bench, SWE-bench Verified,SWE-bench Multimodal, etc, but with special emphasis on covering eight of the most widely used programming languages.

Issues were collected from GitHub, cleaned, verified and manually controlled, resulting in a set of 2132 issues.

The benchmark is evaluated using 12 different LLMs on three code-agentic systems and assessed in terms of resolve rate.

Analysis is carried out including performance vs issue difficulty, length of the fix, type of issue, cross-file fix patches and cost.

**Dataset Code Accessibility:**

Yes

**Dataset Code Comments:**

dataset, webpage and leaderboard are published and reachable.

**Ethical Considerations:**

No, there are no or only very minor ethics concerns

**Final Justification:**

Strong paper.

Other reviewers seem to agree for acceptance.

**Limitations Weaknesses:**

No obvious weaknesses were identified.

**Strengths Contributions:**

The work is solid. It addresses the limitation of most code benchmarks that cover only one of a handful of programming languages.
The construction of the dataset is solid. The experimentation seems appropriate, it covers a reasonable subset of both open and proprietary LLMs, on three different agentic frameworks.

Top configuration in terms of performance reaches 21%, giving space for future improvements and preventing early saturation. Experiments also vast discrepancy of performance across languages justifying the introduction of this kind of benchmarks.
The analysis is appropriate going deeper on the essential aspects this kind of benchmark needs as listed above:  issue difficulty, issue type, fix length, etc.

Authors also did a good job covering recent work and mentioning similar, contemporary, the latter covering still less programming languages and having fewer issues to solve.

---

> ### Author Rebuttal · Authors · 2025-07-28
>
> We sincerely appreciate your high recognition of Multi-SWE-bench. Your positive feedback would serve as a strong motivation for us to further maintain and develop Multi-SWE-bench.

---

### Decision · Program_Chairs · 2025-09-18

**Decision:**

Accept (poster)

**Comment:**

This submission creates a benchmark for resolving software engineering issues akin to SWE-bench. What sets this dataset apart is in its comprehensive coverage across different programming languages, allowing them to analyze performance generalization across languages, in addition to analysis of difficulty and cross-file issues. There was no disagreement that the paper should be accepted, as all reviewers agreed this work constituted a reasonable expansion of SWE-bench with respect to these axes, with clear areas for future improvement.